# A proteogenomic analysis of clear cell renal cell carcinoma in a Chinese population

Yuanyuan Qu[1,2,6], Jinwen Feng [1,6], Xiaohui Wu[1,6], Lin Bai[1,6], Wenhao Xu[1,2,6], Lingli Zhu[1,6], Yang Liu [1], Fujiang Xu [1], Xuan Zhang[1], Guojian Yang[1], Jiacheng Lv[1], Xiuping Chen[1], Guo-Hai Shi[1,2], Hong-Kai Wang[1,2], Da-Long Cao[1,2], Hang Xiang[1], Lingling Li[1], Subei Tan [1], Hua-Lei Gan[2,3], Meng-Hong Sun[2,3], Jiange Qiu[4], Hailiang Zhang[1,2✉], Jian-Yuan Zhao [1,5✉], Dingwei Ye[1,2✉] & Chen Ding [1✉]

Clear cell renal cell carcinoma (ccRCC) is a common and aggressive subtype of renal cancer. Here we conduct a comprehensive proteogenomic analysis of 232 tumor and adjacent non-tumor tissue pairs from Chinese ccRCC patients. By comparing with tumor adjacent tissues, we find that ccRCC shows extensive metabolic dysregulation and an enhanced immune response. Molecular subtyping classifies ccRCC tumors into three subtypes (GP1–3), among which the most aggressive GP1 exhibits the strongest immune phenotype, increased metastasis, and metabolic imbalance, linking the multi-omics-derived phenotypes to clinical outcomes of ccRCC. Nicotinamide N-methyltransferase (NNMT), a one-carbon metabolic enzyme, is identified as a potential marker of ccRCC and a drug target for GP1. We demonstrate that NNMT induces DNA-dependent protein kinase catalytic subunit (DNA-PKcs) homocysteinylation, increases DNA repair, and promotes ccRCC tumor growth. This study provides insights into the biological underpinnings and prognosis assessment of ccRCC, revealing targetable metabolic vulnerabilities.

[1] Department of Urology, Fudan University Shanghai Cancer Center, State Key Laboratory of Genetic Engineering, Collaborative Innovation Center for Genetics and Development, School of Life Sciences, Institute of Biomedical Sciences, and Human Phenome Institute, Fudan University, Shanghai, China. [2] Department of Oncology, Shanghai Medical College, Shanghai, China. [3] Tissue Bank & Department of Pathology, Fudan University Shanghai Cancer Center, Shanghai, China. [4] Cell Signaling and Proteomics Research Center, Academy of Medical Science, Zhengzhou University, Zhengzhou, China. [5] Institute for Developmental and Regenerative Cardiovascular Medicine, MOE-Shanghai Key Laboratory of Children's Environmental Health, Xinhua Hospital, Shanghai Jiao Tong University School of Medicine, Shanghai, China. [6] These authors contributed equally: Yuanyuan Qu, Jinwen Feng, Xiaohui Wu, Lin Bai, Wenhao Xu, Lingli Zhu. ✉email: zhanghl918@163.com; zhaojy@fudan.edu.cn; dwyeli@163.com; chend@fudan.edu.cn

Renal cell carcinoma (RCC) is among the top 10 malignant carcinomas[1]. Clear cell (cc)RCC, accounting for ~75% of RCC cases, is an aggressive histological RCC subtype. Various high-throughput genomics studies have revealed global somatic alteration patterns in ccRCC and their association with clinical outcomes[2–4]. Loss of chromosome 3p, resulting in inactivation of various tumor suppressor genes (VHL, PBRM1, BAP1 and SETD2), has been defined as the earliest driver event in ccRCC[2,3]. VHL is the substrate recognition component of an E3 ligase complex, regulating HIF1α and HIF2α by ubiquitin-proteasome system. Loss of VHL leads to the aberrant accumulation of HIF proteins, which in turn results in constitutive activation of glycolysis and angiogenesis. However, apart from VHL, function of other hot-spot genes on 3p were not fully studied, which impede the understanding of the 3p loss in the ccRCC.

Multiomics strategies encompassing genome and expression profiling of multiple tumor types have elucidated novel molecular subtypes and abnormally activated signaling pathways, as well as potential therapeutic targets[5–22]. The Cancer Genome Atlas (TCGA) and Clinical Proteomic Tumor Analysis Consortium (CPTAC) have published landmark multiomics studies[3,7], improving our cognition of ccRCC. The TCGA study presented the most comprehensive genomic analysis of ccRCC and demonstrated the association between aggressive cancers and metabolic reprogramming[3]. The CPTAC conducted an integrated proteogenomics analysis in 103 ccRCC cases, which revealed the tumor-specific proteomic/phosphoproteomic alterations and the immune signature of ccRCC[7]. However, these previous ccRCC multiomics studies were mostly based on patients in Western populations. More importantly, the survival differences of RCC patients receiving targeted therapy between races was reported[23]. In consideration of the ethnic and geographic genetic differences between Western and Eastern populations in ccRCC[24,25], comprehensive proteogenomics studies of ccRCC in Eastern populations are in urgent need. Moreover, approximately 30% of early stage ccRCC patients eventually develop recurrence or metastasis, which highlights the necessity and great potential to explore the underlying molecular features of disease progression and biomarkers for monitoring in early stage ccRCC using multiomics data.

Metabolic reprogramming is a cancer hallmark and presents opportunities for cancer diagnostics, prognostics, and therapeutics[26], which is also observed in ccRCC[27]. Morphologically, ccRCC cells are lipid- and glycogen-laden, implicating altered fatty acid and glucose metabolism in the development of ccRCC. In addition, metabolic reprogramming in ccRCC was well studied at the levels of metabolome and transcriptome[3,28]. The TCGA analysis of ccRCC highlighted the prognostic values of the transcript levels of metabolic enzymes[3]. However, it was reported that transcriptome and metabolome showed discordance[28]. The recent CPTAC study also reported the uncoupling of oxidative phosphorylation related mRNA and protein expression in tumors[7]. As proteins are the direct executors of metabolic reaction, these results indicated that it was necessary to portray the metabolic reprogramming in ccRCC using proteome data.

In this study, we conduct genomic and proteomic profiling of 232 paired tumor (T) and tumor adjacent (TA) samples of Chinese ccRCC patients with a median follow-up of 85 months (range, 3–138 months). We find that 3p loss and 12q gain are the most important arm-level CNA events influence overall survival (OS) and progression free survival (PFS), respectively. Moreover, our study reveals two major features of ccRCC tumors, dysregulated metabolism and immune, and corresponding aberrant transcription factor (TF) activities. Integrated data analysis discloses distinct proteomic subtypes of ccRCC, connecting the genetic aberrations, proteomic features and clinical outcomes of ccRCC. Further, we identify NNMT as biomarker for poor prognosis, and verify that NNMT overexpression mediated homocysteine metabolism dysregulation is a potential therapeutic opportunity for renal cell carcinoma.

## Results

**Proteogenomic landscape of Chinese ccRCC.** We collected 232 paired tumor and adjacent non-tumor tissues from Chinese ccRCC patients (with an age range 17–84) based on strict criteria (Supplementary Fig. 1a) and conducted proteogenomic analysis (Fig. 1a). Clinicopathological indicators, including sex, clinical manifestation, laterality, tumor size, chronic diseases status, tumor node metastasis (TNM) stage, and International Society of Urological Pathology (ISUP) grading classification are summarized in Supplementary Data 1. Each tumor/adjacent sample was checked by an expert pathologist to confirm the sample quality according to the following standards: (1) histopathologically defined ccRCC tumors; (2) tumor samples with tumor cell rate (tumor purity) > 90%; (3) no tumor cells in the adjacent tissues (Supplementary Fig. 1a). Freshly frozen tissues were used for proteomics analysis and whole-exome sequencing (WES). WES was conducted in 224 paired samples; samples from 8 patients were excluded due to low DNA quality.

WES data of tumor adjacent tissues were used as a reference to detect genetic variants in ccRCC. The mean sequencing coverage in the hg38 reference genome was 120.5× for tumor tissues and 68.72× for adjacent tissues (Supplementary Data 1, Supplementary Fig. 1b). Among the 224 sample pairs, 10,475 non-silent mutations in 6,875 genes and 1,203 silent mutations were detected. VHL was the most frequently mutated gene in this cohort (64.3%), followed by PBRM1 (24.5%), BAP1 (10.7%) and SETD2 (8.9%) (Fig. 1b), consistent with previous studies[2,3,24,25] (Fig. 1c). Interestingly, mutation frequencies of these genes exhibited ethnic and geographic variations among the East Asian[2,24,29], TCGA[3], and European[25] cohorts. Specifically, VHL (45.2–64.3% in East Asian vs. 46.2% in TCGA vs. 73.4% in European) and SETD2 (8.0–12.2% in East Asian vs. 13.9% in TCGA vs. 19.1% in European) had the highest mutation frequency in the European cohort. In contrast, PBRM1 (24.5–34.2% in East Asian vs. 42.1% in TCGA vs. 39.4% in European), and MTOR (4.5–6.6% in East Asian vs. 9.2% in TCGA vs. 8.5% in European) had highest mutation frequencies in the TCGA cohort (Fig. 1c). Mutational spectra revealed that C > T transversion (27.0%) was the dominant mutation in the Chinese and TCGA cohorts (Supplementary Fig. 1c). The frequency of T > A transversions was higher in the Chinese cohort than in the TCGA cohort (21.0% vs. 10.6%, Supplementary Fig. 1c). When we decomposed the mutation spectra using the Catalogue of Somatic Mutations in Cancer (COSMIC) database[30], five single-base substitution (SBS) signatures (SBS1, SBS5, SBS22, SBS40, SBS52) were detected (Supplementary Fig. 1d). Signatures SBS1, SBS5, and SBS40 were considered to be correlated with patient age. SBS22 was associated with exposure to aristolochic acid (AA), a Chinese herbal ingredient associated with renal injury and ccRCC carcinogenesis[24,25,31], corroborating that AA exposure is a carcinogenic factor for Chinese ccRCC. Moreover, patients with the AA signature showed higher mutational burden (t test, p = 0.00028, Supplementary Fig. 1e), but no significant difference on survival, compared with patients without AA signature (Supplementary Fig. 1f).

For proteomic data analysis, Spearman's correlation coefficient was calculated for all quality control (QC) runs using HEK293T cell samples (Supplementary Fig. 1g). The average correlation coefficient of the QC samples was 0.95 (range, 0.82–0.99), demonstrating consistent stability of the mass spectrometry (MS)

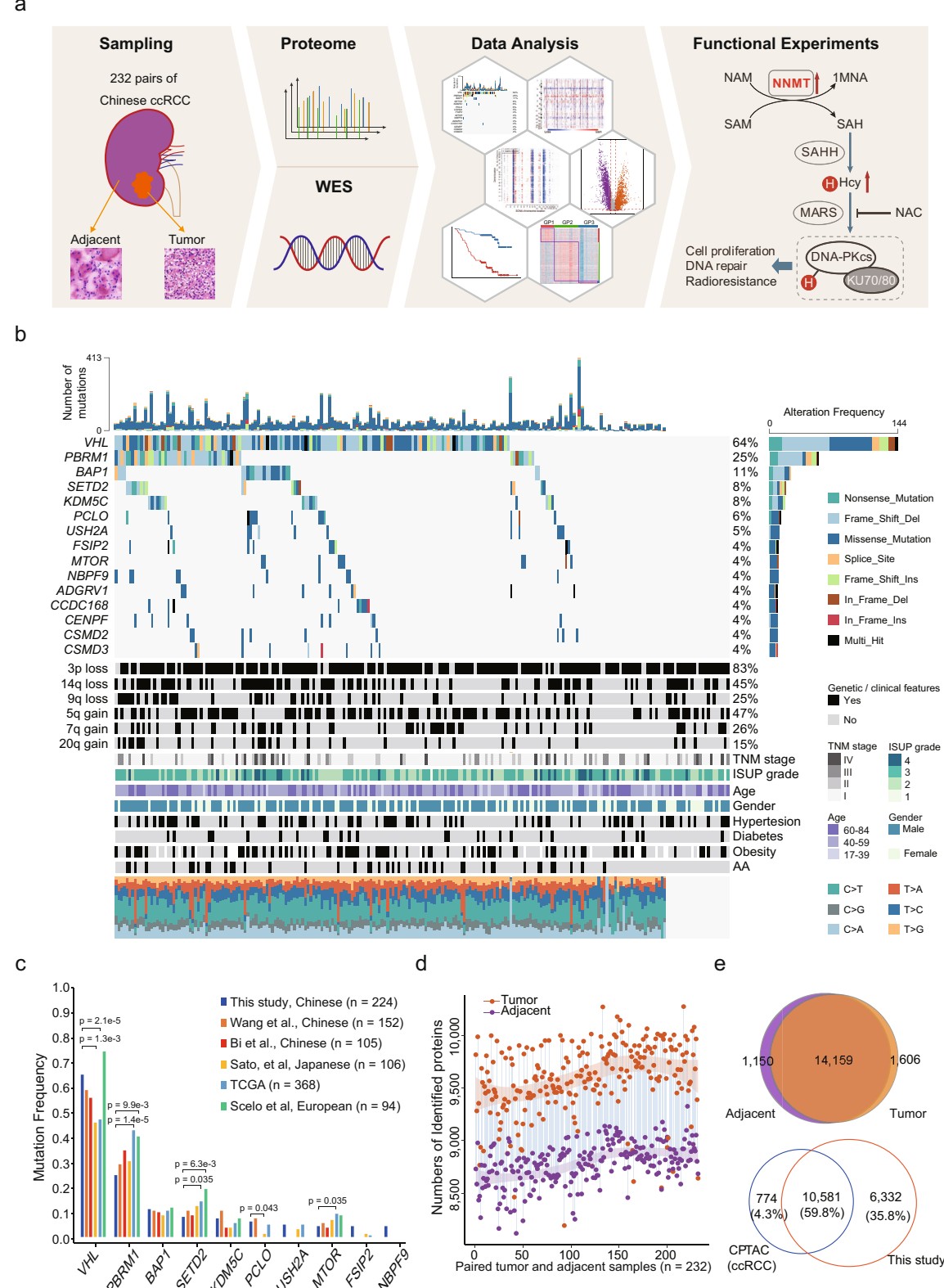

**Fig. 1 Proteogenomic Landscape of Chinese ccRCC. a** Schematic representation of the multiomics analyses of ccRCC, including sample preparation, protein identification, WES, and function verification. **b** Genomic profile and associated clinical features of 224 ccRCC patients. **c** Comparison of frequently mutated genes among Chinese, Japanese, European, and TCGA cohorts. *P* values derived from two-sided Fisher's exact test. **d** Overview of proteomic profiles of pairwise ccRCC samples. The dashed curves fitted by lasso regression show the distribution of protein identifications. The shading that underlies the lasso curves denotes the 95% confidence intervals. **e** The upper Venn diagram shows the overlap of proteins identified in tumors and adjacent normal tissues. The lower Venn diagram shows that proteins identified in this study cover most of the proteins identified in the CPTAC ccRCC cohort.

platform. The correlation coefficients of the 232 tumor and adjacent samples were 0.35–0.89 (mean, 0.78) and 0.71–0.91 (mean, 0.84), respectively (Supplementary Fig. 1h, i). The lower correlation values among tumor samples indicated a high level of tumor heterogeneity. We identified 8,119–10,273 proteins in each sample (Fig. 1d). Significantly more proteins were identified in tumors (median, 9,608) than in paired adjacent tissues (median, 8,807) (two-sided paired $t$ test, $p < 2.2e-16$, Supplementary Fig. 1j), indicating the complexity of tumor micro-environment. In total, 16,915 proteins were detected in the 232 paired samples (Supplementary Fig. 1k), among which 14,159 proteins were in common between tumor and adjacent tissues, whereas 1,606 and 1,150 proteins were detected specifically in tumor and adjacent tissues, respectively (Fig. 1e). When we compared our data with those of the CPTAC ccRCC study[7], 10,581 proteins were found in both cohorts, whereas 6,332 and 774 proteins were detected specifically in the Chinese cohort and the CPTAC cohort, respectively (Fig. 1e). Proteome quantification was conducted using the iBAQ algorithm, followed by normalization to fraction of total (FOT) as reported previously[32,33] (Supplementary Fig. 1l). In summary, this study provided a comprehensive landscape of Chinese ccRCC at both the genomic and the proteomic levels.

**Genomic alterations and their proteomic consequences.** Somatic (S)CNAs in Chinese ccRCC were identified using GISTIC. The most frequently arm-level deleted chromosomal regions in our cohort were 3p (83%), 14q (44%), and 9q (25%), whereas the most frequently arm-level amplified regions were 5q (49%), 7q (26%), 5p (24%), and 7p (24%) (Fig. 2a, Supplementary Data 2), which is consistent with previous reports. SCNA profiles were shown in Supplementary Fig. 2a; 3p 25.1 loss was found in 91% of tumors. We used Cox regression to identify associations between arm-level CNAs and clinical outcomes (Fig. 2b). Loss of 3p was associated with better survival (hazard ratio [HR] = 0.47; 95% confidence interval [CI], 0.24–0.907; $p = 0.021$), whereas 9p loss (HR = 2.2; 95%CI, 1.18–4.02; $p = 0.0133$), 9q loss (HR = 1.9; 95%CI, 1.01–3.45; $p = 0.0472$), 14q loss (HR = 1.9; 95%CI, 1.04–3.47; $p = 0.0359$), and 12q gain (HR = 2.2; 95%CI, 11.1–4.52; $p = 0.0256$) were associated with poorer survival (Fig. 2b, Supplementary Data 2). Multivariate Cox regression analysis of these CNA events showed that 3p loss was the most significant events associated with overall survival (Supplementary Data 2, Supplementary Fig. 2b). More interestingly, we found that 3p copy number (CN) was also associated with overall survival in dosage (Fig. 2c). Based on 3p CN, we divided 3p loss ccRCC into two groups, low burden (LB) and high burden (HB). LB 3p loss tumors were significantly enriched in higher TNM stage, comparing with HB 3p loss tumors (Fisher's exact test, $p = 0.042$, Supplementary Fig. 2c). Consistently, HB 3p loss tumors had better OS than LB 3p loss tumors, which also observed in the TCGA cohort (Fig. 2d). In addition, *BAP1* was mutated more frequently in LB 3p loss tumors than in HB 3p loss tumors (Fisher's exact test, $p = 0.024$), whereas *PBRM1* had a significantly lower mutation frequency in LB 3p loss tumors than in HB 3p loss tumors (Fisher's exact test, $p = 0.019$, Supplementary Fig. 2c). This phenomenon was also found in the TCGA dataset ($p < 0.05$, Supplementary Data 2).

Genomic alterations that affect gene expression levels at the same locus are said to act in *cis*, whereas an impact of another locus is defined as a *trans*-effect[10,19]. Diagonal patterns in Fig. 2e represent *cis*-effects of CNAs, and vertical patterns indicate *trans*-effects. The CNAs with *cis*-effect were centered around 3q, 5q, 12p, and 12p, whereas those with *trans*-effect were centered around 3p, 3q, 9p, 9q, 14p, and 14q (Fig. 2e). Among CNAs with

a strong *trans*-effect, 3p CN was negatively correlated with overall survival (Fig. 2c).

We further investigated the dosage cascade regulation of 3p CN in 3p loss ccRCC. The results of *trans*-effect analysis revealed that 67 proteins were positively correlated with 3p CN (Spearman's correlation, $q < 0.05$), converged on complement and coagulation cascades (Supplementary Fig. 2d, Supplementary Data 2). Meanwhile, 1,343 proteins were negatively correlated with 3p CN, enriched in pathways including glycolysis, metabolism of lipids and signaling by receptor tyrosine kinases (Supplementary Fig. 2d, Supplementary Data 2). Consistently, gene set enrichment analysis (GSEA) revealed that gene sets related to complement and coagulation cascades and epithelial mesenchymal transition (EMT) were enriched in LB 3p loss tumors, and glycolysis/gluconeogenesis was enriched in HB 3p loss tumors (Fig. 2f). To further investigate the direct gene targets located at 3p, we prioritized genes showed significant *cis*-effect in both this cohort and the CPTAC cohort at proteome level and evaluated their prognostic powers (Fig. 2g). Six genes (*SLC4A7*, *PRKCD*, *CDCP1*, *NBEAL2*, *HIGD1A*, and *STT3B*) showed significant *cis*-cascade regulation and correlation with survival (Fig. 2g). High expression of SLC4A7 (Solute Carrier Family 4 Member 7) was associated with poor survival in this cohort, which was also observed in the TCGA cohort at mRNA level (Fig. 2h). SLC4A7, also known as NBCn1, plays an important role in cellular net acid extrusion and intracellular pH (pHi) balance. Previous study reported that expression of SLC4A7 was sensitive to pHi decrease[34]. We examined the correlation of SLC4A7 protein expression level with lactate abundance across 317 cell lines from Cancer Cell Line Encyclopedia (CCLE)[35]. The result showed a significant positive correlation between SLC4A7 level and lactate abundance (Spearman's correlation = 0.38, $p = 7e-14$, Fig. 2i), indicating the importance of SLC4A7 to cellular pH homeostasis in tumor cells.

It was noted that 3p loss would lead to enhanced glycolysis (Fig. 2f, Supplementary Fig. 2d) and more acidic waste product. Combined with the impaired SLC4A7 expression, 3p loss ccRCC tumors might have lower pHi. It was reported a decreased pHi limited the capacities of proliferation, migration, and invasion of tumors[36]. Consistently, EMT was enriched in the LB 3p loss tumors (Fig. 2f). Moreover, we found that the activities of SMAD3 and SMAD4 were decreased in the HB 3p loss group than in the LB 3p loss groups (Fig. 2j). Moreover, SMAD3 and SMAD4 activities, highly correlated with TGF-β signaling and EMT, were negatively correlated the expressions of epithelial cell-cell junction markers and positively corelated with the expressions of mesenchymal proteins and migration associated proteins (Fig. 2k), indicating the latent association of TGF-β–SMAD–EMT axis in ccRCC. In conclusion, ccRCC with a higher degree of 3p loss exhibited increased glycolysis and impeded net acid extrusion, which might result in pHi decreased. The decreased pHi in tumor cells might result in the suppression of EMT by the TGF-β–SMAD–EMT axis, which led to good prognosis for patients (Fig. 2l).

**Disease progression-associated Proteogenomic alterations in ccRCC.** To further identify associations between arm-level CNAs and disease progression, univariate analysis of PFS was performed by Cox regression method. We found that gains of chromosome 7q, and 12q, losses of 8p, 9p, and 9q were associated with poorer PFS (Fig. 3a, Supplementary Data 2). The results of Cox regression multivariate analysis further revealed 12q gain was the most significant event associated with PFS (Fig. 3b, Supplementary Data 2).

We identified a *cis*-cascade of Nucleosome Assembly Protein 1 Like 1 (NAP1L1) at 12q, which was also observed in the CPTAC

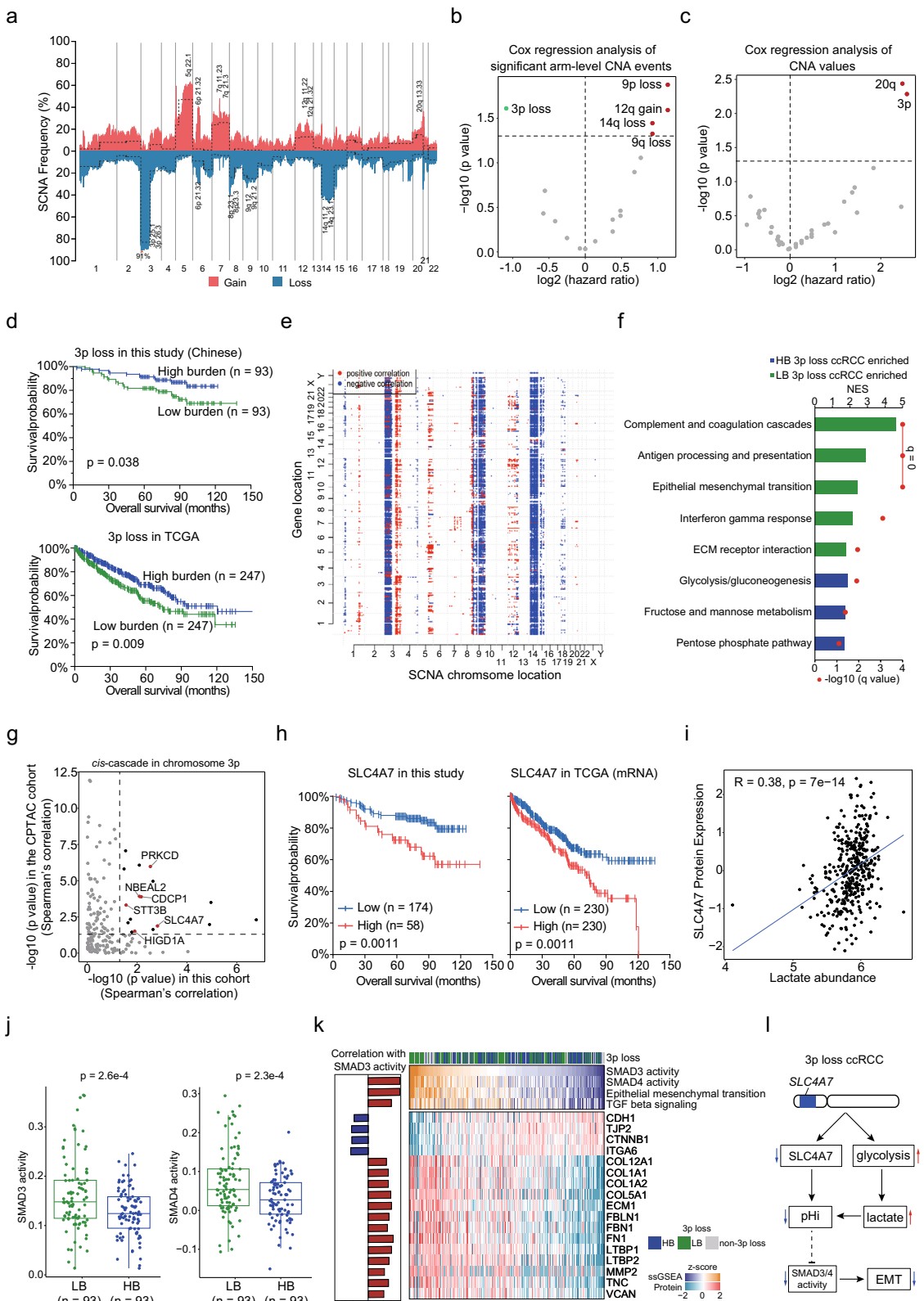

KIRC cohort (Spearman's correlation < 0.05, Fig. 3c) and the TCGA cohort (Supplementary Fig. 3a). Moreover, higher expression of NAP1L1 was associated with shorter PFS in both this study (log-rank test, $p = 0.00047$, Fig. 3d) and the TCGA cohort (Supplementary Fig. 3b). NAP1L1 were reported to promote cell proliferation by inhibiting the expression of CDKN1C[37]. Consistently, we observed the negative correlation of NAP1L1 and

CDKN1C in both our cohort and the CPTAC cohort (Fig. 3e). To further confirm the relationship of NAP1L1 and cell proliferation, we correlated NAP1L1 expression levels with multi-gene proliferation scores (MGPS)[38]. The result showed that NAP1L1 levels were positively correlated with MKI67 expression level and MGPS (Fig. 3f, Supplementary Fig. 3c, d). In addition, we found high-proliferation tumors (IHC [immunohistochemistry], MKI67

**Fig. 2 Profiles of CNAs and Effects of CNA on Somatic Mutations, Proteome, and Overall survival. a** Frequency of SCNAs. Copy number gains and losses are indicated in red and blue, respectively. The dotted line indicates the frequency of arm-level CNA events. **b, c** Cox regression analysis of significant arm-level CNA events and CN. **d** Kaplan–Meier curves of OS for patients with different 3p loss burden in the Chinese and TCGA cohorts (two-sided log-rank test). **e** Correlations of CNA (x axes) with protein abundance (y axes). Significant ($q < 0.01$) positive (red) and negative (blue) correlations are shown. **f** GSEA of patients with LB 3p loss ($n = 93$) compared to patients with HB 3p loss ($n = 93$). NES, normalized enrichment score. **g** Prioritizing genes in chromosome 3p. Chromosome 3p gene encoded proteins, with prognostic values (HR > 1, $p < 0.05$), were annotated by red. **h** Kaplan–Meier curves of PFS for patients with different SLC4A7 abundances in the Chinese and TCGA cohorts (two-sided log-rank test). **i** The correlation between SLC4A7 protein expression and lactate abundance (two-sided Spearman's correlation test) ($n = 370$). **j** Comparison of SMAD3 and SMAD4 activities between LB 3p loss group ($n = 93$) and HB 3p loss group ($n = 93$). P values are derived from two-sided t test. Boxplots show the median (central line), the 25–75% interquartile range (IQR) (box limits), the ±1.5×IQR (whiskers). **k** Pathways and proteins, involved in EMT, significantly associated with SMAD3 or SMAD4 activities. The left panel shows Spearman's correlation between SMAD3 activities and pathway scores/protein abundances. **l** Proposed model of the pH imbalance induced EMT impairment in 3p loss ccRCC.

positive ≥10%)[39] expressed more NAP1L1 than low-proliferation tumors (IHC, MKI67 positive < 10%) (Fig. 3g). More interestingly, NAP1L1 levels were positively correlated with levels of multiple cancer stem cell (CSC) markers (Fig. 3e), indicating high expression of NAP1L1 might be associated with ccRCC dedifferentiation. In conclusion, we supposed that 12q gain increased NAP1L1 expression and further promoted cell proliferation, resulting in rapid progression of disease (Fig. 3h). To optimize treatment options for these patients, we correlated the well-established mRNA signatures Angiogenesis, T-effector, and Myeloid inflammation[40] with NAP1L1 levels using TCGA mRNA data. We found that tumor T-effector scores were positively correlated with NAP1L1 levels (Supplementary Fig. 3f). Moreover, we found that tumors with 12q gain had higher T-effector scores while lower Angiogenesis scores (Supplementary Fig. 3g), indicating these patients might be more likely to benefit from anti-angiogenesis combined with immune checkpoint blockade (ICB) therapy rather than anti-angiogenesis therapy.

Although early stage (stage I&II) ccRCC patients who received nephrectomy have a high 5-year survival rate of 95%, ~30% of these patients eventually developed recurrence or metastasis[41]. We found that 12q gains were enriched in short PFS group ($n = 38$, PFS < 5 years, SP group) compared with long PFS group ($n = 149$, PFS ≥ 5 years, LP group) (Fisher's exact test, $p = 0.031$, odds ratio = 3.27) (Fig. 3i). Consistently, NAP1L1 showed higher expression in the SP group (Supplementary Fig. 3h). GSEA showed that pathways related to immune response and oncogenic signaling pathways, such as MTORC1 signaling, and MYC targets were upregulated in SP group (Fig. 3j, k). On the contrary, cell adhesion related pathways were upregulated in LP group (Fig. 3j, k). Pathway scores of MTORC1 signaling and tight junction showed corresponding differences between SP and LP groups and significant association with PFS (Fig. 3l). In addition, among proteins significantly upregulated in SP group (FC > 2, $p < 0.05$), 20 proteins which could be detected in plasma (annotated by the human protein atlas [HPA] database[42]) were significantly correlated with clinical outcomes (Fig. 3m). We used these 20 proteins to distinguish between SP and LP patients, which achieved a high accuracy, with the area under the receiver operating characteristic (AUROC) of 0.87 (Fig. 3n). The robustness of these proteins for prognostic prediction needs to be confirmed in further studies.

**Proteomic alterations in ccRCC compared to adjacent tissues revealing tumorigenic changes and biomarker candidates**. To obtain a general insight into the proteomic alterations in ccRCC tumor tissues compared to adjacent tissues, 7,267 proteins detected in >25% of the patients were further analyzed (Supplementary Data 3). Principle component analysis (PCA) and hierarchical clustering analysis revealed a clear distinction

between the proteomes of tumor and adjacent tissues (Fig. 4a, Supplementary Fig. 4a). PCA distances among tumor tissues were significantly lower than those among tumor adjacent tissues, corroborating tumor heterogeneity. In total, 3,187 differentially expressed proteins (DEPs) were identified in tumor tissues compared with adjacent tissues (Benjamini–Hochberg-adjusted $p < 0.01$, two-sided paired t test, FC > 2), including 1,719 downregulated and 1,468 upregulated proteins (Fig. 4b, Supplementary Data 3). Previous studies revealed that ccRCC originated from the proximal tubule epithelial cells[21,28]. We observed that eight proximal tubule-specific proteins (GGT1, BHMT, LRP2, DPYS, AGMAT, SLC22A8, SLC22A13, and HRSP12), annotated by the HPA database[42], were significantly downregulated in tumor tissues (Supplementary Fig. 4b), revealing the loss of tissue identity of ccRCC tumors. In addition, we found that patients with lower expression of proximal tubule-specific proteins, GGT1 and BHMT in tumors appeared to have poorer clinical outcomes (log-rank test, $p < 0.05$) (Supplementary Fig. 4c).

We performed pathway enrichment analysis of the cellular process alterations in ccRCC. Tumor-upregulated proteins significantly converged on pathways including glycolysis/gluconeogenesis (e.g., HK2, PFKP, ALDOA, PGK1), interferon gamma mediated signaling (e.g., OAS1, FCGR1A, TRIM5, GBP1), immune response (e.g., IFITM3, CD40, HLA-DMA), antigen processing and presentation (e.g., CANX, PSME2, B2M), ECM-receptor interaction (e.g., COL2A1, VWF, LAMA4), NF-κB signaling (e.g., BCL2, LCK, LYN, NFKB2), HIF-1 signaling (e.g., EGFR, FLT1, HK2, PDK1, PIK3CD), and PI3K-AKT signaling (e.g., GYS2, ITGA5, PDK1, PIK3CD, TCR2). Proteins, downregulated in tumor tissues, were mainly involved in pathways related to kidney functions (collecting duct acid secretion, and proximal tubule bicarbonate reclamation), and PPAR signaling (e.g., ACOX2, FABP3, CPT2, EHHADH, RXRA), citrate cycle (e.g., CS, FH, IDH2, ACO2, SDHA), oxidative phosphorylation (e.g., ATPs, NDUFABs, NDs), biosynthesis of amino acids (e.g., DDC, AFMID, PSAT1, GLS), fatty acid degradation (e.g., HADH, EHHADH, ACAT1), anion transmembrane transport (e.g., SLC22A6, ABCC2, SLC22A7), and xenobiotic metabolic process (e.g., EPHX1, UGT1A9, BPHL) (Fig. 4c).

Kidney is a metabolic organ. As ccRCC is characterized by aberrant metabolic pathways that control energetics and biosynthesis, it is important to learn how metabolic bioprocesses are altered at the proteome level in ccRCC. To this end, we evaluated the activities of metabolism-related pathways using single sample (ss)GSEA[43] (Supplementary Fig. 4d, Supplementary Data 3). Glycogen metabolism, and glycolysis were upregulated in tumor tissues. In contrast, most metabolic pathways, including tricarboxylic acid (TCA) cycle, oxidative phosphorylation (OXPHOS), amino acid metabolism, lipids metabolism, one-carbon metabolism, and metabolism of vitamins and cofactors were downregulated (Supplementary Fig. 4d). Correspondingly,

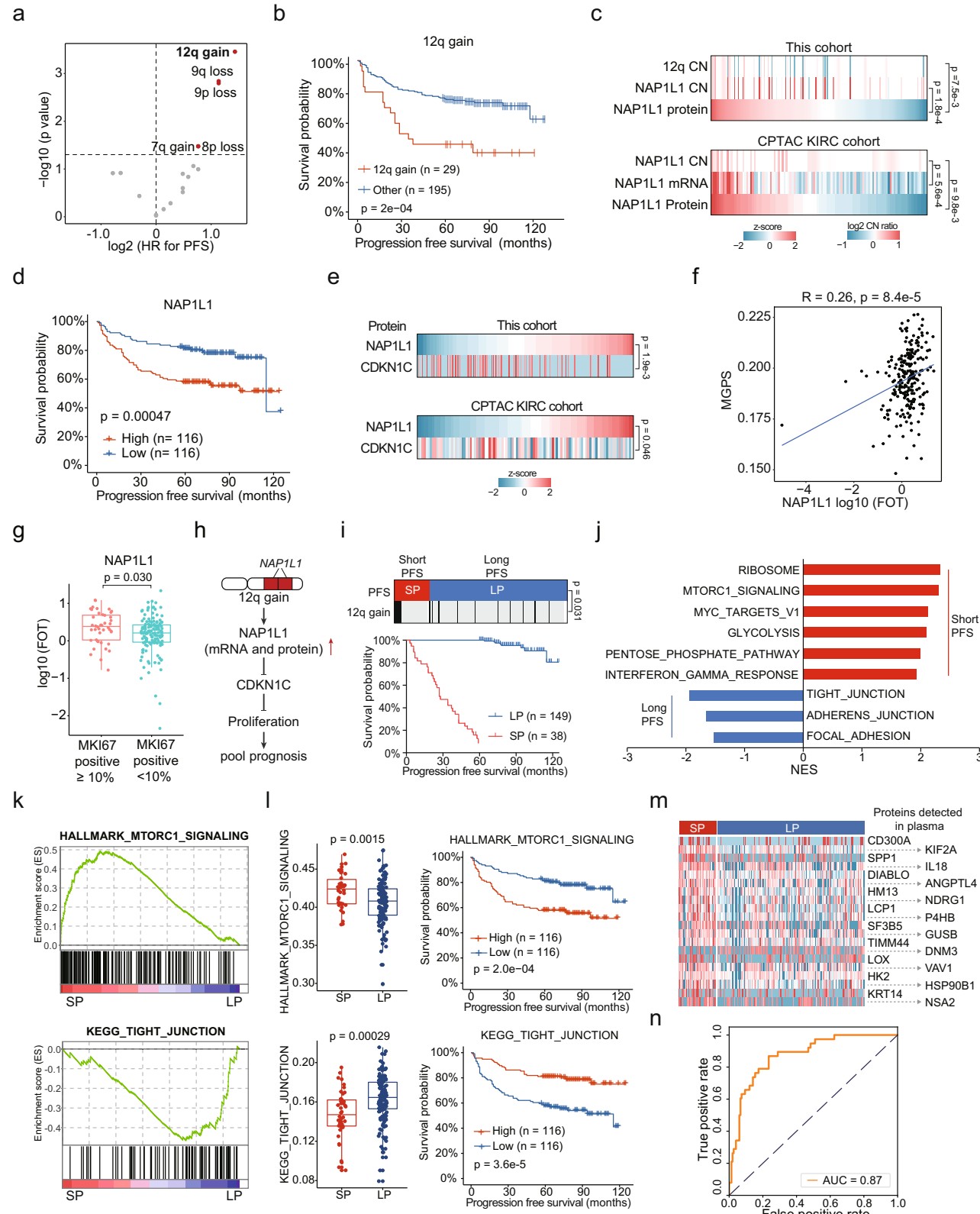

only 177 metabolism-related proteins were upregulated in tumor tissues, comparing with 538 downregulated metabolism-related proteins (Supplementary Data 3), consistent with previous studies that ccRCC was characterized by downregulation of most metabolic bioprocesses[32,33].

Previous studies had disclosed the Warburg effect in ccRCC, which was also observed in our data. The upregulated SLC2A1

(GLUT1) (T/TA = 9.49), HK2 (T/TA = 36.69), PFKP (T/TA = 15.97), PKM (T/TA = 4.08), and LDHA (T/TA = 8.58) suggested the increased of glucose utilization for lactate fermentation in ccRCC tumor vs. adjacent tissues (Fig. 4d, Supplementary Fig. 4e). The "clear cell" morphology was a canonical phenotypic feature of ccRCC, which was associated with lipid accumulation in the cytosol. Our proteome data showed decreased levels of enzymes

**Fig. 3 Proteomic Alterations in ccRCC Compared to Adjacent Tissues. a** Cox regression analysis of significant arm-level CNA events for PFS.
**b** Kaplan–Meier curves of PFS for patients with or without 12q gain (two-sided log-rank test). **c** Cis-effect of 12q gain on NAP1L1 in this study and CPTAC
cohort. P values are derived from two-sided Spearman's correlation test. **d** Kaplan–Meier curves of PFS for patients with different NAP1L1 abundances (two-
sided log-rank test). **e** Negative correlations between NAP1L1 and CDKN1C abundances in this study and CPTAC cohort. P values are derived from two-sided
Spearman's correlation test. **f** Positive correlations between NAP1L1 abundances and MGPS scores in this study (two-sided Spearman's correlation test).
**g** Comparison of NAP1L1 abundances between tumors with different MKI67 IHC results (MKI67 positive ≥ 10%, $n = 44$; MKI67 positive < 10%, $n = 139$).
P value is derived from two-sided t test. Boxplots show the median (central line), the 25–75% IQR (box limits), the ±1.5×IQR (whiskers). **h** Proposed model
explaining the 12q gain-induced disease progression in ccRCC. **i** Up, 12q gains were enriched in SP group compared with LP group. P value is derived from
two-sided Fisher's exact test. Down, Kaplan–Meier curves of PFS for SP group and LP group (two-sided log-rank test). **j** GSEA of SP group patients compared
with LP group patients. NES, normalized enrichment score. **k** Enrichment plots of MTORC1 signaling in the SP group and tight junction in the LP group.
**l** Comparison of MTORC1 signaling and tight junction scores between the SP ($n = 38$) and LP ($n = 149$) groups and the associations of MTORC1 signaling
and tight junction scores with PFS. P values are derived from two-sided t test. Boxplots show the median (central line), the 25–75% IQR (box limits), the
±1.5×IQR (whiskers). **m** Heatmap of plasma proteins significantly upregulated in SP group than LP group. Higher expressions of these proteins were
associated with shorter PFS. **n** The area under the receiver operating characteristic (AUROC) of the 20 plasma proteins predictor.

involved in β-oxidation (ECHS1, HADH, HADHA, EHHADH,
ACAT1) in ccRCC. In addition, ACLY, a key enzyme of de novo
fatty acid synthesis, was significantly upregulated in ccRCC
tumors (Fig. 4d, Supplementary Fig. 4e). Differences of these
enzymes between tumor and adjacent tissues indicated the
potential mechanism of lipid accumulation and "clear cell"
morphology in ccRCC.

More interestingly, we found distinct one-carbon metabolism
imbalance in ccRCC (Fig. 4d, e, Supplementary Fig. 4e). By
surveying the CPTAC data[7], we found that uncoupling of mRNA
and protein level was not only observed in OXPHOS, but also in
one-carbon metabolism (Supplementary Fig. 4f). One-carbon
metabolism, which supports various bioprocesses, including
nucleotide biosynthesis, amino acid homeostasis, epigenetic
maintenance, and redox defense, plays central roles in carcino-
genesis and tumor progression[37]. Loss of SHMT1 (T/TA = 0.19)
and ALDH1L1 (T/TA = 0.22) would attenuate formate clearance.
Overexpression of MTHFD1L (T/TA = 4.37) and MTHFD2
(T/TA = 6.89, identified in 52 samples) might in turn result in
the increased formate generation (Fig. 4d and Supplementary
Fig. 4e). NNMT (T/TA = 17.37) and DNMT1 (T/TA = 3.97),
upstream enzymes of homocysteine (Hcy) metabolism, were
overexpressed in tumors, leading to enhanced Hcy generation.
Hcy can be removed through catabolic processes via different
enzymes, such as methionine synthase (MTR), betaine-Hcy S-
methyltransferase (BHMT, T/TA = 0.20; BHMT2, T/TA = 0.17),
and cystathionine-beta-synthase[38]. In tumor tissues, we observed
impairment of the cytosolic one-carbon cycle (ALDH1L,
MTHFD1, and MTHFR) (Fig. 4d and Supplementary Fig. 4e),
limiting generation of $CH_3$-THF, coenzyme of MTR, which was
confirmed by the metabolome data[28]. Thus, the increased Hcy
production enzymes and decreased Hcy clearance enzymes
indicated the accumulation of Hcy in ccRCC, which was also
observed in the MSK ccRCC metabolite cohort[28]. To further
confirm this result, we examined the levels of Hcy metabolites in
tumor and adjacent tissues. Hcy was 2.7-fold more concentrated
in ccRCC tumors than in adjacent tissues (two-sided paired t test,
$p = 1.9e$-13) (Fig. 4f). More excitingly, by investigating the
enzymes involved in formate metabolism, we found that patients
with higher expression of ALDH1L1 and SHMT1 appeared to
have better prognostic outcomes, whereas patients with higher
expression of MTHFD1L appeared to have poorer prognostic
outcomes (log-rank test, $p = 0.059$, Fig. 4e). As for enzymes
involved in Hcy metabolism, higher expressions of NNMT and
DNMT1 were associated with poorer prognosis, whereas higher
expressions of BHMT and BHMT2 were associated with better
prognosis (log-rank test, $p < 0.05$, Fig. 4e). Together, our study
interrogated concrete protein expression alterations in one-
carbon metabolism in ccRCC, highlighting the significance of

one-carbon metabolism dysregulation during ccRCC pathogen-
esis and development.

A list of transcription factors are overactivated in most human
cancer cells, which makes them targets for the development of
anticancer drugs[44]. Among 3,187 DEPs between tumor and
adjacent tissues, we found 49 TFs showed increased expressions
in tumor tissues, and 9 TFs showed decreased expressions. To
further evaluate the differences of transcriptional factor activities
between ccRCC tumor and adjacent tissues, we performed
ssGSEA using TF target genes from the DoRothEA[45] as the gene
sets (Supplementary Data 3). We found that STAT1, STAT2,
NKFB2 showed both increased protein expressions and activities
in ccRCC tumors, and HNF4A showed both decreased protein
expressions and activities (Fig. 4g). Higher expressions of STAT1
and STAT2 were associated with poorer survival (Fig. 4h). We
further analyzed the tumor-upregulated target proteins of STAT1
and STAT2, and found these target proteins were mainly
enriched in interferon gamma signaling, and response to immune
(Fig. 4i). The tumor-downregulated target proteins of HNF4A, a
key transcription factor that drives proximal tubule differentia-
tion, were enriched in proximal tubule bicarbonate reclamation,
anion transmembrane transport, and xenobiotic metabolic
process (Fig. 4i).

Deep proteogenomic characterization of ccRCC tumors and
adjacent tissues also provided a powerful dataset to nominate
candidate biomarkers. Using a stringent cutoff for quantitative
difference and consistency (T/TA > 10 in more than 80% of
paired samples), we identified 27 potential biomarkers upregu-
lated in tumor tissues (Fig. 4j). Among the 27 proteins, three
(FCGR1A, ITGAX, and MSR1) are cluster of differentiation (CD)
markers, three (NNMT, FCGR1A, and ALOX5) can be targeted
by FDA-approved drugs, eight (NNMT, CA9, SLC2A3, IL4I1,
INPP5D, PLIN2, ALOX5, and SLC16A3) are metabolism-related
proteins. Among the 26 proteins with IHC staining data in the
HPA dataset, we found that CA9, a common used renal caner
biomarker[46,47], showed medium to strong tumor-specific staining
in only 50% renal cancer samples (Fig. 4k). NNMT, PLOD2,
HAPLN3, PLIN2, and SLC16A3 showed medium to strong
tumor-specific staining in more than 90% renal cancer samples,
indicating higher general applicability of these biomarkers
(Fig. 4k). Taken all together, the distinct and consistent
differences between ccRCC tumor and tumor adjacent might
have high potential utility in elucidation of mechanism, early
diagnosis, and prognosis stratification.

**Proteomic subtypes of Chinese ccRCC.** TNM staging of ccRCC
reflects tumor size, position, lymph node involvement, and
metastasis. ISUP grading of ccRCC is based on tumor differ-
entiation and morphology[48]. Both TNM staging and ISUP

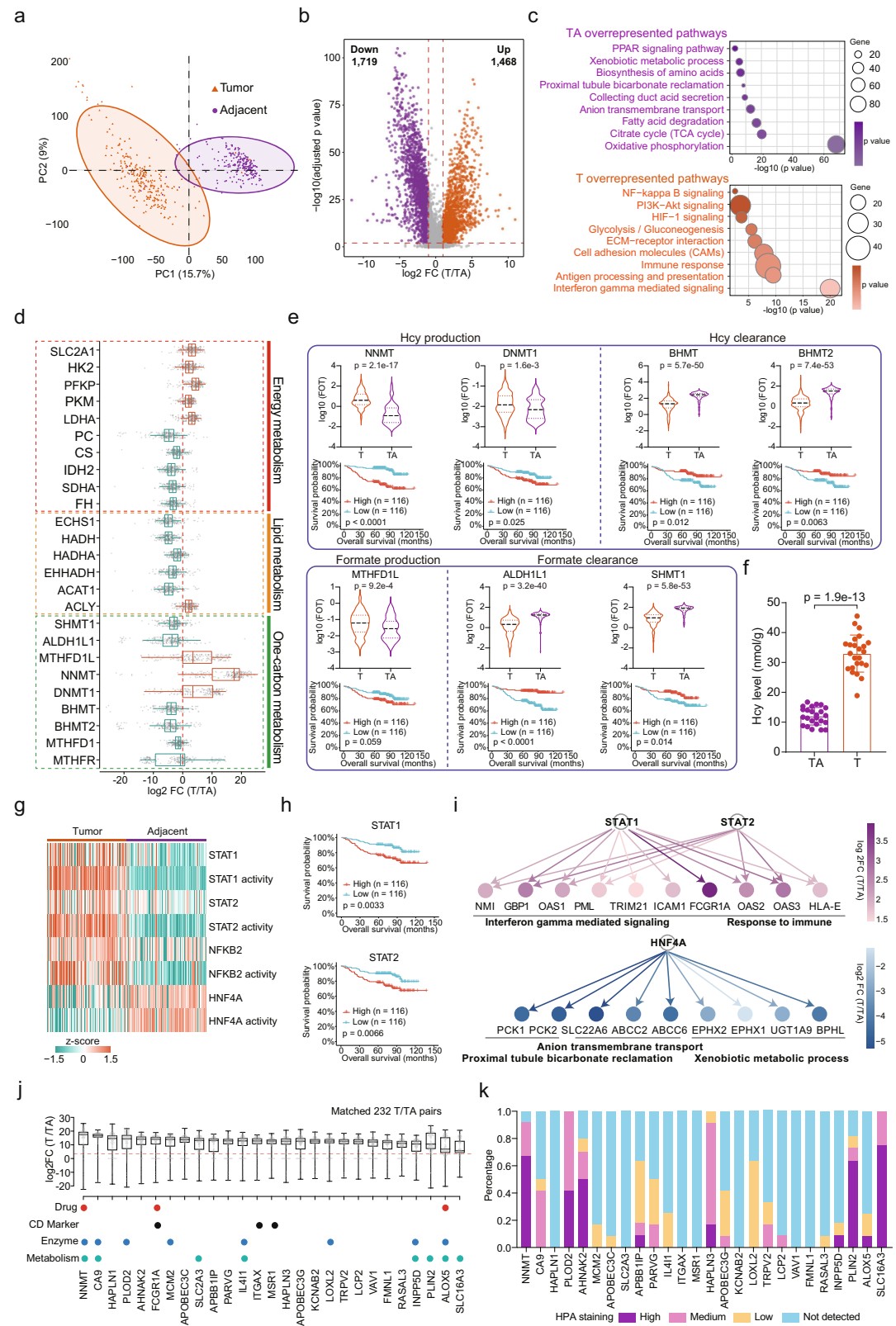

grading were associated with clinical outcomes in this cohort (Supplementary Fig. 5a, b). We conducted separate differential analyses to reveal proteomic differences in different stages and grades. Compared with stages I&II ccRCC, stages III&IV ccRCC displayed elevated expression of complement and coagulation cascades (FGA, PLG), neutrophil degranulation (CEACAM8, CD177), membrane trafficking (KIF2A, SRC), and translation

(EIF4E, EEF1A1) (Supplementary Fig. 5c, Supplementary Data 4). Consistently, complement and coagulation cascades, neutrophil degranulation, membrane trafficking, and translation were upregulated in high-grade vs. low-grade tumors (Supplementary Fig. 5d, Supplementary Data 4).

Given the inter-tumoral heterogeneity, it is important to perform molecular subtyping. We employed consensus

**Fig. 4 Proteomic Alterations in ccRCC Compared to Adjacent Tissues Reveal Tumorigenic Changes and Biomarker Candidates. a** PCA of 7,267 proteins in 232 paired tumor and adjacent tissue samples. Orange, tumor tissue; purple, tumor adjacent tissue. **b** Volcano plot showing DEPs (two-sided paired $t$ test, Benjamini–Hochberg-adjusted $p$ value < 0.01, FC > 2) in tumor and adjacent tissues. Proteins that were significantly overexpressed in tumor/adjacent tissues are presented with orange/purple filled scatters. **c** DEPs in tumors and adjacent tissues, and their associated biological pathways. **d** Dysregulation of metabolic bioprocesses in ccRCC. Alterations of representative proteins depicted as log2 FC (T/TA) ($n = 232$). Boxplots show the median (central line), the 25–75% IQR (box limits), the ±1.5×IQR (whiskers). **e**, Differentially expressed one-carbon metabolic enzymes between tumor and adjacent tissues (two-sided $t$ test) and their associations with clinical outcomes (two-sided log-rank test). **f** Hcy concentrations in tumor and adjacent tissues ($n = 24$). $P$ values are derived from two-sided paired $t$ test. Data are shown as mean ± SD. **g** Transcription factors showed both increased/decreased protein expressions and activities in ccRCC tumors. **h** The increased activity of STAT1 and STAT2 in ccRCC and their association with prognosis (two-sided log-rank test). **i** Regulatory networks of the TFs and their downstream target proteins. **j** Abundance fold changes (FCs) for selected highly elevated proteins annotated with potential clinical utilities ($n = 232$). Drug (FDA-approved drug target), CD marker, and enzyme were annotated by HPA. Metabolism (metabolism-related protein) was annotated by Reactome. Boxplots show the median (central line), the 25–75% IQR (box limits), the min–max (whiskers). **k** "HPA staining proportions" indicate the proportion of ccRCC sections staining positive for the specific marker in the HPA database.

---

clustering[49] to identify ccRCC proteomic subtypes. ccRCC was classified into three subtypes, GP1, GP2 and GP3, comprising 55, 99, and 78 cases, respectively (Fig. 5a, Supplementary Fig. 5e, f). Remarkably, the proteomic subtypes significantly differed in OS (log-rank test, $p < 0.001$, Fig. 5b, c) and PFS (log-rank test, $p < 0.001$, Supplementary Fig. 5g, h). Among the three subtypes, GP1 showed a particularly high mortality risk (HR = 7.8; 95% CI, 4.33–14.1; $p = 9.23e-12$, Supplementary Data 1) and 80% GP1 cases eventually developed progressive disease (Supplementary Fig. 5g). The results of multivariable analysis after adjusting for TNM stage and ISUP grade authenticated subtype GP1 as an independent prognostic factor (HR, 3.15; 95% CI, 1.59–6.21; $p = 9.6e-4$; Supplementary Data 1). After patient stratification according to TNM stage, proteomics subtypes were still significantly associated with patient survival, regardless of tumor stage (log-rank test, $p < 0.0001$), supporting the superior prognostic power of our proteomic subtyping (Fig. 5c, Supplementary Fig. 5h). To confirm the association between proteomic subtyping and clinical outcomes, we extracted the 20 most representative proteins of each proteome subtype in the Chinese ccRCC cohort to classify CPTAC data[7] into three subtypes (Supplementary Fig. 6a, b). Consistently, CPTAC-GP1 was significantly associated with poorer survival than CPTAC-GP2 and CPTAC-GP3 (log-rank test, $p = 0.001$, Supplementary Fig. 6c), in line with the survival differences among the three subtypes in the Chinese cohort (Fig. 5b), indicating the robustness of proteomic subtyping based on the Chinese ccRCC cohort.

When we separately decomposed the mutation spectra of the three proteome subtypes, we found that 7 patients in GP1 contained SBS23, which was not found in the overall mutation signatures. Significant arm-level CNA events varied among the three subtypes, reflecting profound genomic effects on the ccRCC proteome. Multiple arm-level CNA events, including 8p loss, 9p loss, 9q loss, 18q loss, 12q gain, and 20q gain, were aggregated in GP1 (Fisher's exact test, $p < 0.05$, Fig. 5a, e), accounting for 37.3%, 47.1%, 49.0%, 25.5%, 25.4%, and 31.4% of GP1, respectively. Among these, 9p loss, 9q loss, and 12q gain were risk factors for survival (Fig. 2b). *BAP1* and *ABCA13* mutations, which were associated with poor survival (Supplementary Data 2) occurred more frequently in GP1, accounting for 23.5% and 9.8% of GP1, respectively (Fig. 5f). In contrast, *PBRM1* mutations, mutually exclusive *BAP1* mutations, were enriched in GP2 (Fisher's exact test, $p = 0.038$, Fig. 5f).

We used ESTIMATE[50] to deconvolute tumor microenvironment (TME) compositions (Supplementary Fig. 7a) and conducted overrepresentation analysis of elevated proteins in each subtype (Fig. 5g). In total, 641, 1,838, and 97 proteins were upregulated in GP1, GP2, and GP3, respectively (Supplementary Data 5). GP1 was characterized by a high degree of immune infiltration (Kruskal–Wallis test, $p = 7.9e-16$, Supplementary

Fig. 7a), as indicated by the enrichment of multiple immune-associated pathways, including innate immune system, complement and coagulation cascades, antigen processing-cross presentation, interferon signaling, and T cell receptor (TCR) signaling ($q < 0.05$, Fig. 5g). Consistently, GP1 had the highest immunosuppression, CD8 cluster, and MHC I antigen-presenting machinery (APM) scores (Kruskal–Wallis test, $p < 0.05$, Supplementary Fig. 7a). GP2 displayed high tumor purities (Kruskal–Wallis test, $p < 2.2e-16$, Supplementary Fig. 7a) and increased metabolism-related pathways, including the TCA cycle and respiratory chain, amino acid metabolism, mitochondrial translation, lipid metabolism, and glycolysis/gluconeogenesis ($q < 0.05$, Fig. 5g). GP3 featured the highest stromal scores (Kruskal–Wallis test, $p < 2.2e-16$, Supplementary Fig. 7a), corresponding to upregulation of ECM-related pathways, including ECM organization, collagen formation, elastic fiber formation, and focal adhesion ($q < 0.05$, Fig. 5g). Classification of our proteome data according to established CPTAC subtyping signatures[7] provided further support for the diverse characteristics of the proteome subtypes in the Chinese ccRCC cohort. Specifically, GP1 were mainly CD8[+] inflamed tumors, GP2 were mainly of the metabolic immune-desert subtype, and GP3 were mainly CD8[-] inflamed tumors (Supplementary Fig. 7b). The allocation of CPTAC subtypes in our data was also indicated by the immune and stromal scores (Supplementary Fig. 7a, c).

Given that GP1 represented the most aggressive ccRCC subtype, we investigated detailed molecular expression patterns of GP1 by comparing proteins upregulated in GP1 with those in GP2 and GP3 (FC > 2, $p < 0.05$, Fig. 5h). The molecular characteristics of GP1 tumors were summarized into three categories: immune, metastasis, and metabolic dysregulation. Specifically, proteins involved in TCR signaling (e.g., CD8A, ZAP70, LCK, SYK, PTPRC) and macrophage signatures (e.g., ITGAM, MSR1, FCGR1A, TLR2) were significantly upregulated in GP1. CD163, a marker of tumor-associated macrophages[51,52], was also elevated in GP1 (FC = 2.55, $p = 0.001$). Additionally, inflammasome components (PYCARD, CASP1, CASP4, NAIP, NLRC4, IL18) and inflammation-related molecules (CRP, ALOX5) were highly expressed in GP1. Correspondingly, GP1 had the highest immunosuppression scores among the three subtypes (Kruskal–Wallis test, $p < 0.0001$, Supplementary Fig. 7a). The metastatic potential of GP1 was reflected by the upregulation of ECM remodeling (e.g., MMP9, SPARC, PLOD3, P4HB) and cytoskeleton rearrangement (e.g., VTN, VIM, CFL1, ARPC2). Moreover, GP1 displayed increased expression of angiogenic features (e.g., ANGPTL4, EGFR, AAMP) and complement and coagulation cascade components (e.g., C1, C7, C8, C9, PLG, SERPINE1), which have been associated with aggravated tumor invasion[53,54]. Metabolic dysregulation, including prominent overactivation of the oxidative pentose phosphate (OxPPP)

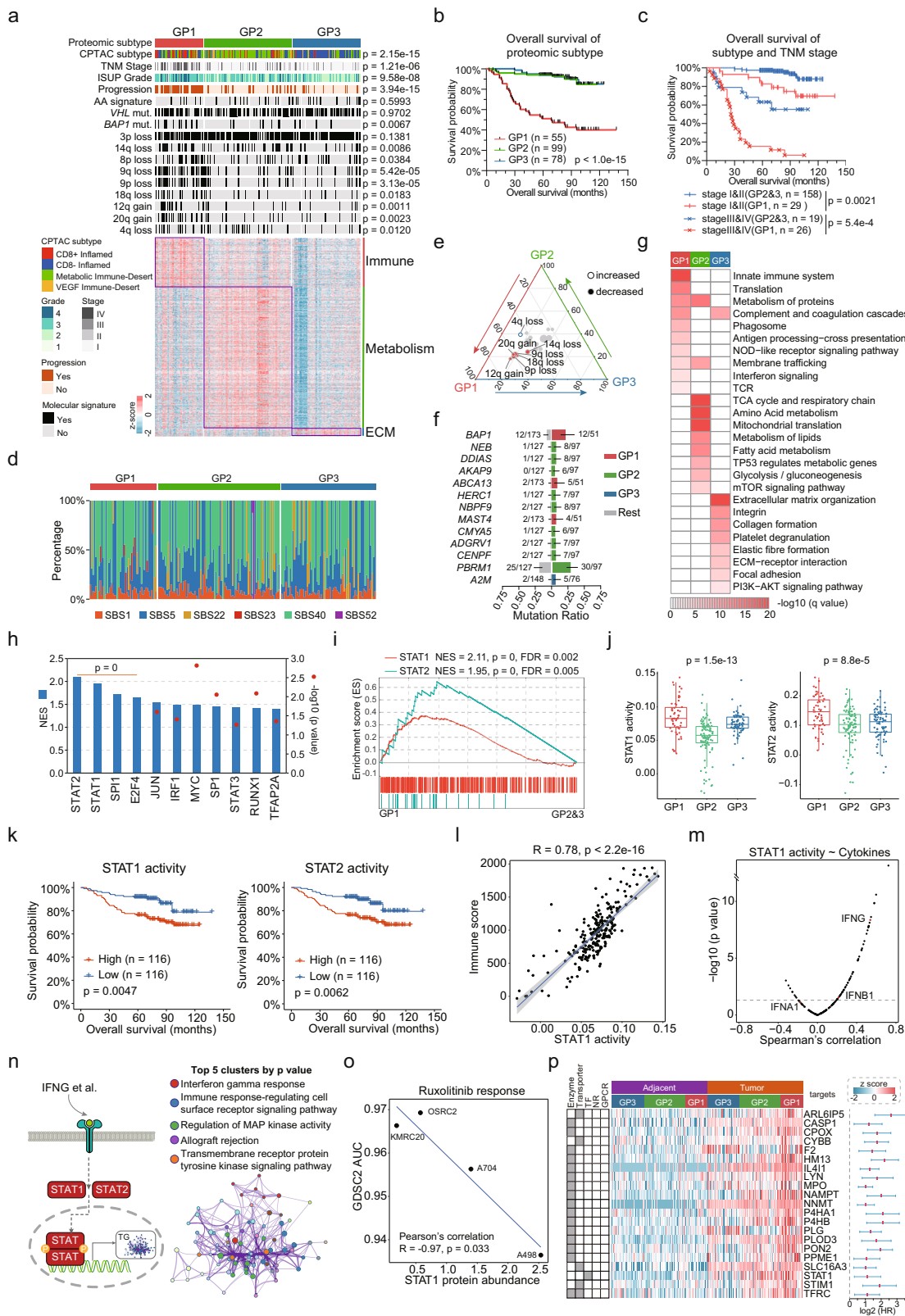

pathway (G6PD, PGD) and dysregulation of Hcy metabolism (NNMT, DNMT1), was another characteristic of GP1 (Supplementary Fig. 7d). In addition, we performed GSEA to screened out the activated TFs in GP1, by using TF targets as gene sets (Fig. 5h, i). Moreover, STAT1 and STAT2, which were defined as ccRCC tumor activated TFs (Fig. 4g), were also significantly upregulated in GP1 ($p < 0.05$, Supplementary Fig. 7e). TF

activities of STAT1 and STAT2, inferred by ssGSEA, further supported this finding (Fig. 5j). As expected, higher STAT1 and STAT2 activities were associated with poorer survival (Fig. 5k). As STAT1/2 played important roles in immune responses, we correlated STAT1/2 activities with immune infiltration in ccRCC. We found that STAT1/2 activities, particularly STAT1 activity (Spearman's correlation, $R = 0.78$, $p < 2.2e-16$), were significantly

**Fig. 5 Proteomic Subtypes of ccRCC and Associations with Genetic Features and Clinical Outcomes. a** Relative abundances of upregulated proteins in the three proteomic subtypes and associations of proteomic subtypes with multiple variables, including CPTAC subtype, TNM stage, ISUP grade, status of progression and genetic features (Fisher's exact test). **b** Kaplan–Meier curves of OS for the three subtypes (two-sided log-rank test). **c** Kaplan–Meier curves of OS for subtypes GP1 and GP2&3 at different TNM stages (stage I&II vs. III&IV) (two-sided log-rank test). **d** Relative percentage of each mutation signatures in the three subtypes. **e** Ternary plot showing the distribution of significant arm-level events in the three subtypes. **f** Genes with differential mutation rates in each subtype (One-sided Fisher's exact test). **g** Upregulated pathways enriched in the three subtypes. **h** Transcription factor activities significantly upregulated in GP1 compared with GP2&3 by GSEA analysis. **i** GSEA plot showing the upregulated STAT1 and STAT2 activities of GP1 tumors. **j** Comparison of ssGSEA inferred activities of STAT1 and STAT2 among three proteomic subtypes (GP1, $n = 55$; GP2, $n = 99$, GP3, $n = 78$). $P$ values are derived from Kruskal–Wallis test. Boxplots show the median (central line), the 25–75% IQR (box limits), the ±1.5×IQR (whiskers). **k** Kaplan–Meier curves of OS for patients with different STAT1 and STAT2 activities (two-sided log-rank test). **l** STAT1 activities were significantly correlated with immune scores (Two-sided Spearman's correlation test). Shaded region indicates 95% confidence interval for the correlation. **m** Correlations between STAT1 activities and cytokine abundances in the CPTAC cohort. **n** Left panel showing the IFN-γ-induced STAT signaling in GP1 ccRCC tumors. Right panel showing Cluster diagram representing pathways enriched by significantly upregulated STAT1/2 targets in GP1 using Metascape (https://metascape.org/). The top 5 clusters by $p$ value are highlighted. **o** STAT1 protein levels are correlated with response to Ruxolitinib across ccRCC cell lines from the GDSC2 (two-sided Pearson's correlation test). **p** Drug target candidates for ccRCC. Left, HPA annotations. Middle, protein abundance. Right, HR for OS of each protein, error bars indicates 95% confidence interval for HR (tumor samples, $n = 232$).

correlated with immune scores in ccRCC (Fig. 5l, Supplementary Fig. 7f), which indicated that activated STAT1 was an essential cause of high immune infiltration in GP1 tumors. To further ascertain the upstream triggers of STAT1, we calculated the correlations between STAT1 activities and cytokine abundances using CPTAC data (Fig. 5m). We found IFN-γ but not IFN-α/β was extracellular signal initiator of STAT signaling (Fig. 5m). Consistently, the defining pathway feature of STAT1/2 target proteins, upregulated in GP1, was interferon gamma response (Fig. 5n). The role of STAT1 in ccRCC was further probed by using cell line perturbation experiments in the Genomics of Drug Sensitivity in Cancer (GDSC) resource[55]. We found that higher STAT1 abundances was correlated with higher drug sensitivity of Ruxolitinib, a JAK-STAT signaling inhibitor (Fig. 5o).

Since GP1 patients had the poorest prognosis and were supposed to be assigned into a clinically high-risk category, they deserved to receive further therapy. As, enzymes, transporters, TFs, nuclear receptors (NRs), and G protein-coupled receptors (GPCRs) were common target for drugs, we further screened therapeutic drug targets in GP1 upregulated proteins based on protein annotation in HPA (Supplementary Fig. 7g). We candidated 21 proteins as potential drug target for ccRCC therapy (Fig. 5p, Supplementary Data 5). STAT1 was the only targetable TF identified using this strategy. Moreover, NNMT, the enzyme play an important role in one-carbon metabolism imbalance in ccRCC, was also identified. In brief, we identified three novel proteomic subtypes of Chinese ccRCC with distinct molecular features that connect the proteomic, genomic, and clinical features of ccRCC.

**NNMT promotes cancer cell proliferation through Hcy accumulation**. We conducted supervised analysis to identify robust and representative prognostic proteins, and we anticipated to screen out drug targets (Fig. 5p). NNMT, an important enzyme in Hcy metabolism, was overexpressed in ccRCC tumors (Fig. 6a) and significantly associated with poor prognosis (Fig. 6b). Furthermore, western blotting (Fig. 6c) and immunohistochemistry (IHC) (Fig. 6d, Supplementary Fig. 8) confirmed that NNMT was overexpressed in ccRCC. Historically, ccRCC has been considered resistant to conventional chemo- and radiotherapy, indicating tolerance to genotoxic stress. Moreover, ccRCC cells are able to proliferate rapidly in a nutrient-depleted microenvironment[27]. Thus, we tested whether high NNMT expression increased the viability of ccRCC under various stresses. NNMT overexpression promoted the proliferation of ACHN and 786-O cells and profoundly enhanced cell proliferation during nutritional stress

or genotoxic stress (Fig. 6e, f). Nutritional and genotoxic stresses induce DNA damage in cells. NNMT overexpression reduced DNA damage in stressed ACHN, 786-O, and 769-P cells as evidenced by the levels of γ-H$_2$AX detected using immunofluorescence staining (Fig. 6g) and western blotting (Fig. 6h) and by DNA damage detection using the comet assay (Fig. 6i), in cultured renal cancer cell lines. These results indicated that NNMT overexpression may contribute to proliferation promotion under stress. NNMT catalyzes methyl transfer from S-adenosyl methionine (SAM) to nicotinamide (NAM) and generates S-adenosyl homocysteine (SAH) and 1-methylenicotinamide (1MNA). Increased NNMT in cells resulted in a decrease in SAM and increases in SAH and 1MNA (Fig. 6j). The level of Hcy, the hydrolysis product of SAH, was also increased significantly (Fig. 6j). Supplementation of SAH or Hcy, but not supplementation of 1MNA or a reduction in SAM through knockdown of MAT, reduced DNA damage (Fig. 6k) and promoted cell proliferation under stress (Fig. 6l). Furthermore, blockade of SAH hydrolysis by knockdown of S-adenosylhomocysteine hydrolase (SAHH, AHCY) in NNMT-overexpressing cells abrogated the DNA damage-reducing effect of NNMT (Fig. 6m, n), suggesting that Hcy, but not SAH, plays a role in the DNA repair- and proliferation-promoting effects of NNMT.

**Lysine homocysteinylation of DNA-PKcs enhances DNA repair**. When present at high levels, intracellular Hcy modifies protein lysine residues, which results in protein lysine-homocysteinylation in cells[56]. We observed increased protein lysine-homocysteinylation (K-Hcy) levels in NNMT-overexpressing (Fig. 7a) and SAH-supplemented cultured renal cancer-derived ACHN, 786-O, 769-P, and A-498 cells (Fig. 7b). Meanwhile, Hcy (Fig. 4f) and K-Hcy (Fig. 6c) levels were increased in ccRCC tumors compared to adjacent tissues. In our previous cell-wide proteomics screen for K-Hcy substrates in HEK293T cells, we observed that DNA-PKcs, a protein required for the non-homologous end-joining pathway of DNA repair, was heavily modified by K-Hcy[56,57]. In ccRCC tumors, we validated that three different lysine residues (K122, K712, and K902) were modified by K-Hcy (Fig. 7c), suggesting that K-Hcy regulates DNA-PKcs-mediated DNA repair. Among the three lysine residues in DNA-PKcs (K122, K712, and K902) that were modified by homocysteinylation, K122 is located within the interface between DNA-PKcs and KU70/KU80, while K712, and K902 are located within the intramolecular interaction region of DNA-PKcs[58] (Fig. 7c). Proteins that physically interact with methionyl-tRNA synthetase (MARS) are more prone to being modified and regulated by K-Hcy[56]. Accordingly, an interaction

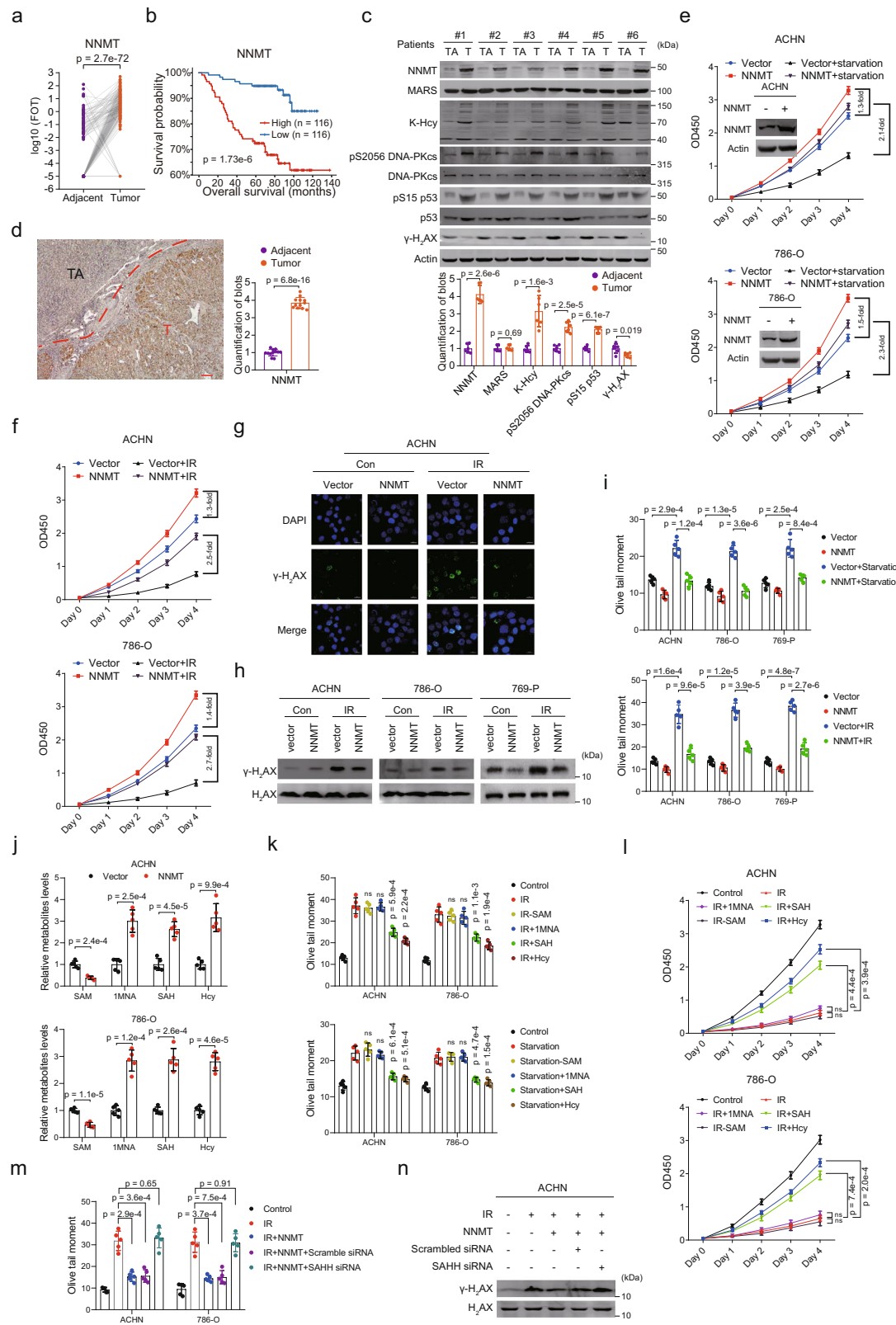

between DNA-PKcs and MARS was confirmed by co-immunoprecipitation assays using either exogenous DNA-PKcs and MARS in ACHN and 769-P cells (Supplementary Fig. 9a), or endogenous DNA-PKcs and MARS in 786-O cells (Fig. 7d). Elevated NNMT, SAH, Hcy, or MARS levels led to dose-dependent increases of DNA-PKcs homocysteinylation in ACHN and 769-P cells (Supplementary Fig. 9b–e). These results confirmed that DNA-PKcs was subject to MARS-mediated K-Hcy modification. Increased NNMT expression resulted in the activation of DNA-PKcs as indicated by increased phosphorylation of DNA-PKcs and its downstream target protein p53 (at Ser15) in ACHN and 769-P cells (Fig. 7e). Moreover, supplementation of either SAH or Hcy or overexpression of MARS induced elevated K-Hcy level of DNA-PKcs and activated the DNA-PKcs pathway in ACHN and 769-P

**Fig. 6 NNMT Promotes Cancer Cell Proliferation through Hcy Accumulation. a** NNMT expression based on our proteomic data. *P* value is derived from two-sided paired t test (*n* = 232). **b** Association between NNMT protein expression and OS (two-sided log-rank test). **c** Upper panel, representative western blots of NNMT, MARS, K-Hcy, DNA-PKcs, DNA-PKcs (pS2056), p53, p53 (pS15) and γ-H₂AX in tumor and tumor tissues. Lower panel, quantified western blot results (*n* = 6). **d** Left panel, IHC results of NNMT expression in tumors and adjacent tissues (scale bars: 50 μm). Right panel, quantified IHC results of 12 sample pairs. TA = tumor adjacent, T = tumor. Results for other samples are shown in Supplementary Fig. 8. **e, f** Cell proliferation associated with various treatments (*n* = 5 repeats per group). **g** Representative plots of immunofluorescence staining of γ-H₂AX in cells under various treatments (scale bars: 20 μm). **h** Western blot analysis of γ-H₂AX and H₂AX in cells under various treatments. **i** Comet assay of DNA damage levels in cells subjected to various treatments. For each group, DNA damage levels in a total of 30 cells from five independent repeats were measured. **j** Relative metabolite levels in cells subjected to various treatments. **k** Comet assay of DNA damage levels in cells subjected to various treatments. **l** Cell proliferation associated with various treatments (*n* = 5 repeats per group). **m** Comet assay of DNA damage levels in cells subjected to various treatments. **n** Western blot analysis of γ-H₂AX and H₂AX in cells under various treatments. Data are shown as mean ± SD in panels **c–f, i–l**. *P* values are derived from two-sided *t* test. Not significant, ns.

cells (Supplementary Fig. 9f–h). In contrast, reducing K-Hcy modification through knockdown of NNMT, SAHH, or MARS inhibited DNA-PKcs activity in ACHN and 769-P cells (Supplementary Fig. 9i–k). Furthermore, phosphorylation levels of DNA-PKcs and p53 were markedly increased in ccRCC tumors vs. adjacent tissues (Fig. 6c). These results were consistent with the comet assay results, which revealed that compared with adjacent tissues, tumors exhibited decreased DNA damage (Fig. 7f) and reduced γ-H₂AX levels (Fig. 6c). The results also indicated that DNA-PKcs is inactivated in ccRCC tumor adjacent tissues and that NNMT-induced hyper lysine-homocysteinylation might promote ccRCC by activating DNA-PKcs.

**Lysine-homocysteinylation facilitates the formation of DNA-PKcs complex.** We next investigated the mechanism by which lysine-homocysteinylation activates DNA-PKcs. DNA-PKcs–KU70/KU80 interaction was significantly enhanced by increased cellular K-Hcy levels induced by NNMT overexpression in ACHN and 769-P cells (Fig. 7g, h). As a result, the activity of DNA-PKcs was enhanced in NNMT-overexpressing cells as determined by monitoring its kinase activity in phosphorylating its substrate p53 in vitro (Fig. 7i) and measuring ADP formation in an ADP-Glo-DNA-PK assay (Fig. 7j). We validated that at increased levels, K-Hcy activates DNA-PKcs, as determined by adding homocysteine thiolactone (HTL) to the in vitro DNA-PKcs assay (Fig. 7k) and measuring intracellular ADP formation in an ADP-Glo-DNA-PK assay (Fig. 7l). Moreover, to mimic the bulky side chain effects of K-Hcy[56], we created two mutant DNA-PKcs constructs in which either Lys122 within the KU70/KU80-binding interface or all three modifiable lysine residues were mutated to tryptophan ("KW" and "3KW" constructs, respectively). Relative to wild-type DNA-PKcs, the mutant DNA-PKcs showed increased binding affinity to KU70/KU80 (Fig. 7m, n) and enhanced DNA-PKcs activity, as determined by an in vitro DNA-PKcs assay (Fig. 7o). In addition, increased NNMT expression in 786-O and ACHN promoted xenograft tumor growth in nude mice, especially in IR-treated cell xenografts, whereas inhibition of K-Hcy by intraperitoneal injection of N-acetyl-cysteine (NAC)[56,57] delayed xenograft growth (Fig. 7p, q). Together, these results confirmed that NNMT upregulation induces hyper K-Hcy, which activates DNA-PKcs and promotes tumor growth (Fig. 7r).

**Discussion**
Our comprehensive proteogenomic study in 232 ccRCC tumor and tumor adjacent tissue pairs of Chinese patients provided insights into ccRCC protein profiles and biology. The genomic profile of ccRCC revealed the somatic mutations and CNAs in Chinese ccRCC patients. Comparison of genome alterations in Chinese and Western ccRCC cohorts emphasized the genetic diversity across geographic regions and revealed features of Chinese ccRCC. Loss of chromosome 3p has long been regarded

as the initial event of ccRCC[59,60]. Our study demonstrated that 3p loss was associated with clinical outcomes in a dosage cascade manner in ccRCC. This dosage cascade manner manifested as downregulation of SLC4A7, a transporter regulating cell pH balance, in 3p loss tumors. It was reported that increased pHi was a permissive signal for cell proliferation and survival, facilitated metabolic adaptation and tumor invasion[36]. Consistently, we found that downregulated SLC4A7 decreased pHi and inhibited EMT by reducing SMADs activities. These findings indicated the pH regulation vulnerability in 3p loss ccRCC. Moreover, we found the degree of 3p loss was associated with the frequencies of driver mutations in *BAP1* and *PBRM1*. Further studies to dissect out the crosstalk between chromosome 3p loss and the driven mutations in ccRCC will promisingly promote our cognitions for the carcinogenesis and development of this disease. In addition, our analysis revealed that chromosome 12q gain was the most important CNA event driving disease progression, which was associated with *NAP1L1* amplification involved cell proliferation. Consistently, we found that increased proliferation and decreased cell adhesion were the common features of early stage ccRCC that eventually developed relapse and metastasis.

Proteomic alterations in tumors compared to adjacent tissues revealed two major features of ccRCC, broad metabolic dysregulation and intensive immune response. Previous studies also reported the metabolic dysregulation in ccRCC at transcriptome and metabolome levels[3,28]. Our results revealed three major metabolic imbalances in ccRCC, including energy metabolism, lipid metabolism and one-carbon metabolism at proteome level. These findings were consistent with the metabolomics data from the MSK ccRCC cohort[28]. Despite enzyme levels could not directly reflect the flux of metabolic reactions, the alterations of metabolic enzymes reflected that cancer cells enabled themselves to rapidly proliferate and survive in conditions of nutrient depletion and hypoxia by reprogramming metabolism. Similarly, metabolic reprogramming also indicated the vulnerability of tumor metabolism, which provided more opportunities for therapy. Further research of these concepts to nephrologists and oncologists will guide clinical trials of therapeutics specifically targeted to tumor metabolism, rather than generally toxic to all proliferating cells. Such novel agents are highly likely to be more effective and to have far fewer adverse effects than existing drugs.

Besides, we found that ccRCC showed intensive immune response, which originated from highly activated STAT1 and STAT2 in ccRCC tissues. Aberrant activities of STAT1 have been implicated in cancer development[61–63]. Particularly in renal carcinoma, increased STAT1 expression was associated with high grade, later stage, large tumor size, and lymph node and distant metastasis[64,65]. In our proteomic data, we constructed downstream regulatory target proteins of STAT1. The targets proteins were mainly enriched in interferon gamma mediated signaling. It's worth noting that the target proteins FCGR1A and NNMT[66],

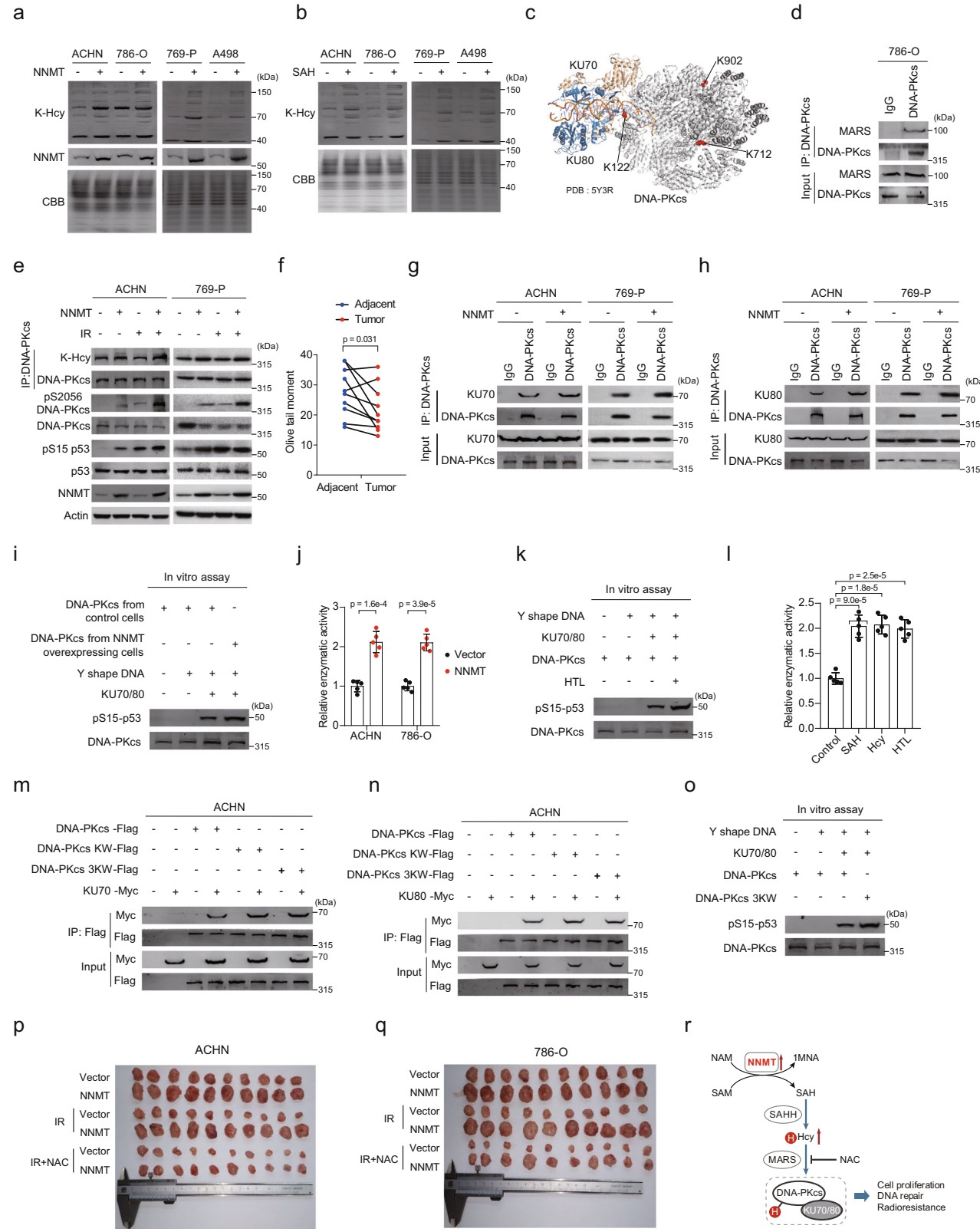

participated in interferon gamma mediated signaling and one-carbon metabolism process, respectively. Notably, NNMT and FCGR1A are potential drug targets for ccRCC therapy in our data.

High immune response is not only the signature of ccRCC, but also associated with more malignant tumors. GP1 exhibited a dominant immune signature, with the highest CD8[+] T cell infiltration and immunosuppression scores (Supplementary Fig. 7a), indicating adaptive immune resistance. We found that IFN-γ-induced STAT signaling responsible for such TME, and STAT1 was the core TF. In addition, GP1 showed higher APM scores (Supplementary Fig. 7a), which reportedly are associated with the immunogenicity of ccRCC tumors[67]. Therefore, we hypothesized that GP1 patients might benefit from immune

**Fig. 7 Lysine Homocysteinylation of DNA-PKcs Increases Cell DNA Repair through Facilitating DNA-PK Complex Formation. a, b** Comparison of K-Hcy levels in cells subjected to various treatments. **c** Structure of DNA-PK complex. K-Hcy sites in DNA-PKcs protein were highlighted in red. **d** Co-immunoprecipitation showing that endogenous DNA-PKcs interacts with endogenous MARS ($n = 3$ biological repeats). **e** Western blot analysis of K-Hcy levels of DNA-PKcs, DNA-PKcs (pS2056), and p53 (Ser15) in cells subjected to various treatments. **f** Comet assay of DNA damage levels in ccRCC tumor vs. adjacent normal tissues ($n = 10$). P value is derived from two-sided paired t test. **g, h** NNMT overexpression enhances the interaction between endogenous DNA-PKcs and endogenous KU70 or KU80 in ACHN and 769-P cells. **i** In vitro DNA-PKcs activity assayed by monitoring its kinase activity in phosphorylating its substrate p53. **j** DNA-PKcs activity indicated by measuring ADP formation in an ADP-Glo-DNA-PK assay ($n = 5$ repeats per group). P values are derived from two-sided t test. Data are shown as the mean ± SD. **k** HTL increases the in vitro DNA-PKcs activity. **l** DNA-PK activity under different treatments ($n = 5$ repeats per group). P values are derived from two-sided t test. Data are shown as the mean ± SD. **m, n** DNA-PKcs KW and 3KW mutants exhibit enhanced binding affinity for KU70 and KU80 compared to wild-type DNA-PKcs. **o** DNA-PKcs 3KW mutant exhibits enhanced kinase activity. **p, q** Tumor size of cell xenografts under different treatments in normal and NAC-administered nude mice. **r** Model depicting NNMT-mediated DNA repair and cell proliferation in ccRCC.

checkpoint inhibitor therapy. Ruxolitinib, targeting JAK-STAT signaling, was commonly used to treat myelofibrosis[68] showed increased sensitivity in ccRCC tumors highly expressed STAT1, suggesting that targeting JAK-STAT signaling might possess of double effects, including inhibiting the tumor growth directly and regulation immune response. It has great potential of targeting JAK-STAT axis for ccRCC treatment beyond anti-VEGF agents.

For the targeted therapeutic strategy, we paid more attention to GP1 patients because they had the poorest prognosis. Twenty druggable candidates were screened out, mainly involving in immune (CASP1, CYBB, F2, MPO, LYN, STAT1), metastasis (P4HA1, P4HB, PLOD3), and metabolism (NAMPT, NNMT, SLC16A3). SLC16A3 (MCT4) transport lactate out of the cell to avoid glycolysis produced intracellular acidosis in ccRCC[69]. Moreover, as mentioned above, we found that loss of chromosome 3p downregulated the expression of SLC4A7 in a *cis*-pattern, which was also associated with net acid extrusion in ccRCC, resulted better survival of patients. These results revealed that ccRCC reprogramed metabolism to meet self-requirement, which also provided the novel therapies. NNMT, a metabolic enzyme, was identified as an important carcinogenic factor and drug target. Some clinical studies using the candidate protein survey strategy[70–74] have reported that NNMT is overexpressed in various tumors, including lung, liver, bladder, colon, and kidney cancers. One proteomics study revealed NNMT as a master metabolic regulator of cancer-associated fibroblasts[75]. In cultured cells, NNMT promotes cancer cell survival, proliferation, migration, and invasion[70–74]. However, the exact oncogenic role of NNMT in ccRCC as well as its metabolic functions in cancer cells have not been determined.

Our recent study demonstrated that Hcy can modify protein lysine residues and turn the metabolic status to cell signaling in colorectal cancer[56]. As NNMT is an upstream metabolic enzyme in Hcy metabolism, we linked NNMT and Hcy in ccRCC tumorigenesis and development. Mechanistically, NNMT overexpression increases Hcy and K-Hcy modification in tumor cells and promoted tumor proliferation. K-Hcy modification of DNA-PKcs, enhancing DNA-PKcs–KU70/KU80 interaction, and finally activates DNA-PK complex. Xenograft experiments revealed that NNMT overexpression empowered the resistance to radiation therapy in renal cell carcinoma. Inhibiting K-Hcy modification by NAC rescued the injuring effect of radiation on tumor. The current study suggested that the NNMT–K-Hcy–DNA-PKcs axis can partially explain the radiotherapy resistance of ccRCC and be considered a potential therapeutic target.

In summary, our study provided a comprehensive proteogenomic landscape of Chinese ccRCC. The dominant pathways that were altered in the ccRCC proteome subtypes revealed the potential molecular mechanism underlying clinical phenotypes and outcomes. We identified a potential druggable protein, NNMT, and demonstrated the value of this multiomics approach.

We believe that this study provides valuable information regarding ccRCC biology and paves the way to novel therapeutic strategies.

## Methods

**Clinical sample collection.** The study was compliant with the ethical standards of Helsinki Declaration II and was approved by the institutional review board of FUSCC (050432-4-1212B). Written informed consent was obtained from each patient before any study-specific investigation was conducted.

We screened 1,556 consecutive patients who underwent radical or partial nephrectomy for the treatment of renal tumors at the Department of Urology of Fudan University Shanghai Cancer Center (FUSCC, Shanghai, China) from January 2007 to March 2014. Electronic medical records were screened retrospectively. In total, 232 eligible ccRCC patients who had undergone radical nephrectomy at the FUSCC were consecutively enrolled. Median follow-up was 85 months (range, 3–138 months). At the last follow-up, 79 patients (34.1%) had progressive disease and 49 patients (21.1%) had died of ccRCC. Clinicopathological indicators, including sex, clinical manifestation, laterality, tumor size, chronic diseases status, TNM stage, and ISUP grading classification are summarized in Supplementary Data 1. Tumor and adjacent non-tumor tissue samples were collected during surgery and are available from the FUSCC tissue bank. Tumor and paired tumor adjacent tissues (collected > 2 cm from the tumor margin) were collected within 30 min after resection, immediately transferred into sterile freezing vials and snap frozen in liquid nitrogen, cut into ~0.5cm$^3$ pieces under $-40$ °C, then split and stored at $-80$ °C until being used. The histologic sections were obtained from top and bottom portions of tumor/adjacent tissues and Hematoxylin and eosin (H&E)-stained for review. Each tumor/adjacent sample was checked by an expert pathologist to confirm the sample quality according to the following standards: (1) histopathologically defined ccRCC tumors; (2) tumor samples with tumor cell rate (tumor purity) > 90%; 3) no tumor cells in the adjacent tissues. H&E-stained slides of tumor and tumor adjacent tissues were uploaded to Figshare (https://doi.org/10.6084/m9.figshare.17206589).

Among the 1,324 excluded patients, 161 patients were diagnosed with benign renal tumor, 118 with urinary tract carcinoma, 326 with non-clear cell RCC, and 89 with other simultaneous or heterochronous malignancies. Further, 577 patients (mainly those who underwent partial nephrectomy) were excluded because of unavailable adjacent normal tissues, and 53 samples failed to pass pathological quality check, such as tumor cell rate < 90% (Supplementary Fig. 1a). All cases were staged according to the 2010 American Joint Committee on Cancer TNM staging system. H&E-stained sections were reviewed by an experienced genitourinary pathologist to determine the ISUP grade, and frozen sections were reviewed to determine the tumor cell rate of the ccRCC tissues.

**DNA extraction and WES.** WES was conducted at Life Healthcare Clinical Laboratory (China). DNA isolated from fresh or frozen tumor tissue samples was used for WES, and matched germline DNA was obtained from adjacent non-tumor tissue samples. DNA was isolated from fresh tissues using DNeasy Blood & Tissue Kit (Qiagen, 69504) according to the manufacturer's instructions. Purified DNA was quantified using a Qubit 3.0 Fluorometer (Life Technologies). For matched germline and tumor tissues, 100 ng of DNA was sheared to 200–300-bp fragments using a Covaris M220 system. Tumor and matched germline DNA libraries were constructed using Accel-NGS 2 S HYB DNA LIBRARY KIT (Swift Biosciences, 23096) and Accel-NGS 2 S MID S1-S4 (Swift Biosciences, 279384). xGen Exome Research Panel v1.0 (IDT, 1056115) and xGen Lockdown reagents (IDT, 1072281) were used for exome enrichment. Dynabeads M-270 Streptavidin (Thermo, 65306) was used for library purification, P5/P7 primers (Nanodigmbio, ND10010) and HotStart ReadyMix (KAPA, KK2612) were used for library amplification. The amplified libraries were purified using SPRISELECT (Beckman, B23319). DNA quality was assessed using a Bioanalyzer High Sensitivity DNA Analysis kit (Agilent Technologies, 5067-4626). Samples underwent paired-end sequencing on a Nextseq CN500 platform (Illumina), with a 150-bp read length. The WES target

region was 33 M. A mean coverage of 100×, a capture rate of 95%, and a dup rate of 40% were achieved for tumor sequencing.

**Somatic variant detection**. Read-depth statistics were calculated using the DepthOfCoverage function in the Genome Analysis Toolkit (GATK v3.8.1.0)[76]. Paired-end reads in Fastq format were aligned to a reference human genome[77] (UCSC Genome Browser, hg38) using Burrows-Wheeler Aligner. Variant calling was conducted following GATK best practices. Somatic single-nucleotide variations and small insertions and deletions were detected using MuTect2 (GATK v4.1.2.0) and were annotated using ANNOVAR[78] based on UCSC known genes. Two longest genes, *TTN* and *MUC16*, were excluded as they tended to acquire numerous mutations by chance in large-scale genome/exome sequencing experiments. The Maftools (v3.10) R package[79] was used to display mutant genes with non-synonymous mutations. MutSigCV[80] was used to identify significantly mutated genes with default parameters. Genes with Benjamini–Hochberg-adjusted $p < 0.01$ were identified as significantly mutated genes.

**Mutation frequency variances across regions**. TCGA ccRCC genome data were downloaded from Xena[81] and data for a European ccRCC cohort were obtained from Scelo et al.[25]. Three East Asian ccRCC genomic cohorts were also collected[2,24,29]. The top 10 most frequently mutated genes in our Chinese cohort and other five cohorts were compared using Fisher's exact test.

**Mutual exclusivity and mutation co-occurrence analysis**. Mutually exclusive or co-occurring sets of genes were detected using the somaticInteractions function in the Maftools R package, using pair-wise Fisher's exact test to detect significant gene pairs. $p < 0.05$ was used as a threshold for statistical significance.

**Mutational signature**. SBSs are defined as a replacement of a certain nucleotide base. There are six possible substitutions: C > A, C > G, C > T, T > A, T > C, and T > G. Considering the nucleotide context, these SBS classes can be further expanded to 96 possible mutation types. The frequencies of the 96 mutation types were estimated for each sample. The non-negative matrix factorization algorithm of SigProfilerExtractor (v1.1)[82] was used to estimate the minimal components that could explain maximum variance among samples. De novo mutation signatures were decomposed using COSMIC v3[30]. After decomposing a matrix of the 96 substitution classes of the samples into five signatures, the contribution of each signature in each sample was estimated.

**CNA calling**. CNAs were called following somatic CNA best practice, using the Calculate Target Coverage function in GATK (v4.1.2.0). We applied Genomic Identification of Significant Targets in Cancer (GISTIC2.0)[83] to identify significantly amplified or deleted focal-level and arm-level events, with q < 0.05 considered significant. The following parameters were used: amplification threshold = 0.1; deletion threshold = 0.1; cap value = 1.5; broad length cutoff = 0.50; remove X-chromosome = 0; confidence level = 0.90; join segment size = 4; arm-level peel off = 1; maximum sample segments = 2,000; gene GISTIC = 1.

**Protein extraction and trypsin digestion**. Collected samples were washed three times with phosphate buffer saline (PBS) buffer to remove blood and debris. Samples were minced and lysed in lysis buffer (8 M urea, 100 mM Tris hydrochloride, pH 8.0) containing protease and phosphatase inhibitors (Thermo Scientific) and then sonicated for 1 min (3 s on and 3 s off, amplitude 25%). The lysates were centrifuged at $14,000 \times g$ for 10 min and supernatants were collected as whole-tissue extracts. Protein concentrations were determined by the Bradford protein assay (TaKaRa, T9310A). Extracts (100 μg protein) were reduced with 10 mM dithiothreitol at 56 °C for 30 min and alkylated with 10 mM iodoacetamide at room temperature in the dark for 30 min. The samples were digested with trypsin using a filter-aided sample preparation method[84]. Tryptic peptides were separated in a home-made reverse-phase C18 column. Peptides were eluted and separated into nine fractions using an acetonitrile gradient (6%, 9%, 12%, 15%, 18%, 21%, 25%, 30%, and 35%) at pH 10. The nine fractions were pooled into three fractions (6%+15%+25%; 9%+18%+30%; 12%+21%+35%), vacuum-dried (Concentrator Plus, Eppendorf), and analyzed by liquid chromatography tandem MS (LC-MS/MS).

**LC-MS/MS**. Samples were analyzed on a Q Exactive HF-X mass spectrometer (Thermo Fisher Scientific) coupled with a high-performance liquid chromatograph (EASY-nLC 1200 System, Thermo Fisher Scientific). Dried peptide samples were dissolved in solvent A (0.1% formic acid in water) and loaded onto a trap column (100 μm × 2 cm, home-made; particle size, 3 μm; pore size, 120 Å; SunChrom) with a maximum pressure of 280 bar using solvent A, then separated on a home-made 150 μm × 12 cm silica microcolumn (particle size, 1.9 μm; pore size, 120 Å; SunChrom) with a gradient of 5–35% mobile phase B (acetonitrile and 0.1% formic acid) at a flow rate of 600 nL/min for 75 min. MS analysis was conducted with one full scan (300–1,400 m/z, R = 120,000 at 200 m/z) at an automatic gain control target of 3e6 ions, followed by up to 20 data-dependent MS/MS scans with higher-energy collision dissociation (target 5e4 ions, max injection time 20 ms, isolation

window 1.6 m/z, normalized collision energy of 27%). Detection was done using Orbitrap (R = 7,500 at 200 m/z). Data were acquired using the Xcalibur software v2.2 (Thermo Fischer Scientific).

**MS platform QC and ccRCC proteome quality assessment**. For QC of MS performance, tryptic digests of HEK293T cell lysates were measured as a QC standard every 2 days. The QC standard was made and run using the same method, conditions, software, and parameters as those used for ccRCC samples. Pairwise Spearman's correlation coefficients were calculated using the R package corrplot (v0.84)[85] for all QC runs, and the results are shown in Supplementary Fig. 1g. The average correlation coefficient among standards was 0.95, with a maximum of 0.82 and minimum of 0.99. Log10-transformed fractions of total (FOTs) for each ccRCC sample (Supplementary Fig. 1h–i) were plotted to show consistency of data quality. The Sva R package v3.34.0[86] was used to evaluate batch effects. We found no significant batch effect in the proteome data. Moreover, PCA plots showed that the batch effects were negligible for batch number, but significant for sample types (Fig. 4a).

**Proteome identification and quantification**. Raw files were processed in Firmiana[33] and searched against the human National Center for Biotechnology Information (NCBI) RefSeq protein database (updated on 04-07-2013, 32,015 entries) using the Mascot 2.4 search engine (Matrix Science Inc). Mass tolerances were 20 ppm for precursor and 50 mmu for product ions. Up to two missed cleavages were allowed. Cysteine carbamidomethylation was set as a fixed modification and methionine N-acetylation and oxidation as variable modifications. Precursor ion score charges were limited to +2, +3, and +4. The data were also searched against a decoy database so that protein identifications were accepted at FDR of 1%. Label-free protein quantifications were calculated using a label-free, intensity-based absolute quantification (iBAQ) approach[32]. Match between runs[87] was used to improve parallelism between tumor/adjacent samples. We built a dynamic regression function based on common peptides in tumor/adjacent samples. Based on the correlation value $R^2$, Firmiana chooses a linear or quadratic function for regression to calculate the retention time (RT) of corresponding hidden peptides and checks the existence of the extracting ion current (XIC) based on the m/z and calculated RT. The program determines the peak area values of existing XICs. We calculated peak area values as parts of corresponding proteins. Proteins with at least 1 unique peptide with a 1% FDR at the peptide level were selected for further analysis. The FOT was used to represent the normalized abundance of a particular protein across samples. FOT was defined as a protein's iBAQ divided by the total iBAQ of all proteins identified in each sample. FOT values were multiplied by $10^5$ for ease of presentation and missing values were assigned $10^{-5}$. Proteome quantification matrix was deposited at Figshare (https://doi.org/10.6084/m9.figshare.17206589).

**Protein and pathway alterations in tumor vs. adjacent tissues**. PCA was conducted to visualize the separation of tumor and tumor-adjacent proteomes using the R package factoextra v1.0.6[88]. In total, 6,111 proteins identified in both > 25% of tumor and tumor-adjacent samples were used for subsequent analysis. Volcano plots were used to display DEPs in tumor and adjacent tissues by applying thresholds of fold change > 2 and Benjamini–Hochberg-adjusted $p < 0.01$. Among the DEPs, 1,719 proteins were significantly upregulated and 1,468 proteins were significantly downregulated in ccRCC tumor tissues. The DEPs were then subjected to KEGG pathway enrichment analyses in DAVID[89], with a $p$ value cutoff of 0.05 (Supplementary Data 3). Protein annotations and signature proteins of the nephrons (including glomerulus, proximal tubule, distal tubule and collecting duct) were obtained from the Human Protein Atlas database (https://www.proteinatlas.org/humanproteome/tissue/kidney).

**Estimate of tumor purity, immune, stromal scores**. Tumor purity, immune, and stromal scores were inferred using the R package ESTIMATE v1.0.11[50]. Although the ESTIMATE algorithm was designed to analyze transcriptome data, some studies have used it for proteome analysis[6,7]. The results indicate the feasibility to evaluate the engagement of each subtype of immune cells.

**Inferring of APM, immunosuppression, CD8 cluster, and pathway scores and TF activities**. APM, immunosuppression, and CD8 cluster signatures were obtained from previous reports[67,90] and computed by ssGSEA[43] using the R package GSVA v1.34.0[43]. Metabolic pathway scores and TF activities for 232 paired ccRCC samples were also computed using the R package GSVA v1.34.0[91] (Supplementary Data 3). KEGG and Reactome gene sets downloaded from the Molecular Signatures Database (MSigDB v7.1, http://software.broadinstitute.org/gsea/msigdb/index.jsp) were set as background. TF target were obtained from DoRothEA (v1.6.0)[45].

**GSEA**. GSEA was conducted using the GSEA 4.0.3 software (http://software.broadinstitute.org/gsea/index.jsp)[92]. KEGG, Reactome, and HALLMARK gene sets downloaded from the MSigDB v7.1 were set as background. FDR < 0.05 was used

as a cutoff. The normalized enrichment score was used to reflect the degree of pathway overrepresentation.

**Associations between clinical characteristics and the ccRCC proteome**. Specific clinical information is presented in Supplementary Data 1. TNM stage- and ISUP grade-specific proteins were screened out based on a fold change > 1.5 and $p < 0.05$. Specific proteins of each TNM stage and ISUP grade were subjected to over-representation analysis using ConsensusPathDB (http://cpdb.molgen.mpg.de/)[93]. Clinical characteristics-associated pathways are listed in Supplementary Data 4.

**Proteomic subtyping of ccRCC, and subtype features**. Consensus clustering was conducted using the R package ConsensusClusterPlus (v.1.52.0)[94] using Pearson correlation as the distance measure. The 1,000 proteins with the highest median absolute deviation in tumor samples were used for k-means clustering with up to five groups. Consensus matrices for k = 2, 3, 4, 5 clusters are shown in Supplementary Fig. 5e–f. The consensus matrix for k = 3 showed clear separation among clusters. The cumulative distribution function of the consensus matrix for each k-value was also measured (Supplementary Fig. 5f). The relative change in area under the cumulative distribution function curve increased by 33% from 2 clusters to 3 clusters, whereas others exhibited no appreciable increase. Thus, proteome clusters were defined using k-means consensus clustering with k = 3. Subtype-specific upregulated proteins are: (1) detected in ≥ 25% tumor samples; (2) expressed higher than other subtypes (FC > 2, two-sided t test, $p < 0.05$). Subtype-specific upregulated proteins were further analyzed in ConsensusPathDB[93]. DEPs of each subtype and relevant enriched pathways are listed in Supplementary Data 5.

**Validation of proteomic subtyping performance**. GSEA was conducted to identify signature proteins of each proteomic subtype using GSEA v4.0.3[92], and the 20 proteins with the highest scores in each subtype were selected. Hierarchical clustering of CPTAC ccRCC cohort[7] (available follow-up is three years at present) proteome data with signature proteins also classified the CPTAC cohort into three subgroups with a similar survival curve in our population, with GP1 showing distinctly worse survival than the other two subtypes (log-rank test, $p = 0.001$) (Supplementary Fig. 6).

**Correlations between subtypes and clinical features**. To evaluate correlations between proteomic subtypes and clinical features, Fisher's exact test was conducted on categorical variables, including driver gene mutations, significant arm-level CNA events, age, sex, hypertension status, obesity status, cardiovascular and cerebrovascular disease status, family history of cancer, TNM stage, ISUP grade, and CPTAC subtype. Only variables that varied significantly among the three proteome subtypes are shown in Fig. 5a. Scaled CPTAC ccRCC proteome data were used to identify signature proteins of each subtype by GSEA. The 20 proteins with the highest GSEA scores were selected as support vectors to build a support vector machine classifier. Chinese ccRCC cohort was divided into four CPTAC subtype using this classifier.

**Effects of CNAs**. Spearman's correlations between CNA values (gene level) and protein abundances were calculated using 14,538 genes quantified at both CNA and proteome levels. CNAs with significant correlation with proteins were selected based on FDR < 0.01. In total, 89,992 CNA and protein pairs showed significant correlation. Correlations were visualized using the R package multiOmicsViz (v1.10.0). Genomic alterations that affect gene expression at the same locus are said to act in *cis* (diagonal patterns in Fig. 2a), whereas an impact of another locus is defined as a *trans*-effect (vertical patterns in Fig. 2a).

**Survival analysis**. The Kaplan–Meier method was used for survival analyses, and groups were compared using the log-rank test. The R survival package 3.2–3[95] and survminer 0.4.8 were used for statistical tests and visualization. The HR was calculated by Cox proportional hazards regression analysis. Variates with $p < 0.05$ were considered to significantly impact prognosis. OS was used as a primary endpoint. Clinical and molecular variates with $p < 0.05$ in single variant analysis were selected for Cox regression multivariate analysis (Supplementary Data 1).

**Drug target analysis**. Target proteins were selected based on three criteria: significantly upregulated in tumor vs. adjacent (Benjamini–Hochberg-adjusted $p < 0.01$, FC > 2), upregulated in GP1 compared to GP2&3 ($p < 0.01$, FC > 2), and associated with poor prognosis (HR ≥ 2, $p < 0.01$). The proteins were mapped using the HPA database (https://www.proteinatlas.org/). Druggable proteins are listed in Supplementary Data 5.

**Cell culture**. Human HEK293T (ATCC, CRL-11268; RRID: CVCL_QW54), A-498 (ATCC, HTB-44; RRID: CVCL_1056) and ACHN (ATCC, CRL-1611; RRID: CVCL_1067) cells were cultured in high-glucose Dulbecco's modified Eagle's medium (DMEM; HyClone) supplemented with 10% fetal bovine serum (FBS; Invitrogen), 100 units/mL penicillin (Invitrogen), and 100 μg/mL streptomycin (Invitrogen). 769-P (ATCC, CRL-1933; RRID: CVCL_1050) and 786-O cells

(ATCC, CRL-1932; RRID: CVCL_1051) were maintained in RPMI 1640 medium (Invitrogen) containing 10% FBS. Cells were incubated in 5% $CO_2$ at 37 °C. Cells were transfected using polyethylenimine (linear, 25 KDa) or Lipofectamine 2000 (Invitrogen). To generate a cell model of nutrition stress, ACHN and 786-O cells were cultured in medium without serum and glucose for 12 h before the assays. To generate a cell model of genotoxic stress, cultured cells were irradiated with 4 Gy X-ray radiation using a linear accelerator (Oncor, Siemens) before the experiments.

**Plasmid construction and transfection**. Whole-length NNMT, MARS, KU70, KU80, and p53 cDNA clones were purchased from Origene. A whole-length DNA-PKcs cDNA clone was obtained from Prof. Yanhui Xu[58]. After confirming the sequences by Sanger sequencing, DNA-PKcs, NNMT, and p53 were amplified and subcloned into the NheI and EcoRI restriction sites of the pcDNA3.1-Flag vector, using ClonExpress MultiS One Step Cloning Kit (#C113-02, Vazyme). KU70 and KU80 were amplified and subcloned into the NheI and EcoRI restriction sites of the pcDNA3.1-Myc vector using the same kit. DNA-PKcs mutants were generated by site-directed mutagenesis using the MutanBEST kit (TaKaRa). The primers used were as follows: NNMT: forward, 5′- ggg aga ccc aag ctg gct agc ATG GAA TCA GGC TTC ACC TCC -3′, and reverse, 5′- tag tcc agt gtg gtg gaa ttc CAG GGG TCT GCT CAG CTT CC -3′; p53: forward, 5′- ggg aga ccc aag ctg gct agc ATG GAG GAG CCG CAG TCA G -3′, and reverse, 5′- tag tcc agt gtg gtg gaa ttc GTC TGA GTC AGG CCC TTC TGT C -3′; KU70: forward, 5′- ggg aga ccc aag ctg gct agc ATG TCA GGG TGG GAG TCA TAT TAC A -3′, and reverse, 5′- tag tcc agt gtg gtg gaa ttc GTC CTG GAA GTG CTT GGT GAG G -3′; KU80: forward, 5′- ggg aga ccc aag ctg gct agc ATG GTG CGG TCG GGG AAT -3′, and reverse, 5′- tag tcc agt gtg gtg gaa ttc TAT CAT GTC CAA TAA ATC GTC CAC A -3′; DNA-PKcs: forward, 5′- ggg aga ccc aag ctg gct agc ATG GCG GGC TCC GGA GCC G -3′, and reverse, 5′- tag tcc agt gtg gtg gaa ttc CAT CCA GGG CTC CCA TCC T -3′; DNA-PKcs K122W: forward, 5′- GA GCT GCT tgg TGT AAA ATT CCA GCC CTG GAC C -3′, and reverse, 5′- TTT ACA cca AGC AGC TCT ATC TTT TGT ATA AAC ACT G -3′; DNA-PKcs K712W: forward, 5′- AA TTT GGC tgg GAG GTG GCA GTT AAA ATG AAG CA -3′, and reverse, 5′- CAC CTC cca GCC AAA TTT CAC AAA TAA AGC AAA -3′; DNA-PKcs K902W: forward, 5′- A GAG ATG tgg CCT GTC ATT TTC CTG GAT GTG TT -3′, and reverse, 5′- T GAC AGG cca CAT CTC TCT AAA GGG CAC TGC AA -3′. For transient transfection, 1 μg of each plasmid was transfected using Lipofectamine 2000 (Invitrogen) according to the manufacturer's instructions.

**RNA interference**. Synthetic oligos were used for siRNA-mediated silencing of *NNMT* (5′-CCTCTCTGCTTGTGAATCCTT-3′), *SAHH* (5′-GTCAGGAGGGCAACATCTTTG-3′), MARS (5′-GGACGGACCUGCUGCUGAATT-3′), and scramble siRNA was used as a control. Cells were transfected with siRNAs using Lipofectamine 2000 according to the manufacturer's protocol. Knockdown efficiency was verified by qRT-PCR or western blotting.

**Gene silencing and overexpression**. For NNMT stable shRNA knockdown or overexpression, cells were co-transfected with pCMV-VSV-G, pCMV-Gag-Pol, and plasmids using the calcium phosphate method[96]. Transfected cells were cultured in DMEM containing 10% FBS for 6 h. Twenty-four hours after transfection, culture supernatant was collected and used for retrovirus preparation to infect cells at 10% confluency in 90-mm-diameter dishes. Cells were re-infected 48 h after the initial infection and selected using 5 μg/mL puromycin (Amresco). NNMT shRNA was cloned into the AgeI and EcoRI restriction sites of the pMKO vector. NNMT was subcloned into the BamHI and EcoRI restriction sites of the pBABE vector using ClonExpress MultiS One Step Cloning Kit. The sequences of primers used were as follows: shNNMT-Forward: 5′- CCG GCC TCT CTG CTT GTG AAT CCT TCT CGA GAA GGA TTC ACA AGC AGA GAG GTT TTT G -3′, shNNMT-Reverse: 5′- AAT TCA AAA ACC TCT CTG CTT GTG AAT CCT TCT CGA GAA GGA TTC ACA AGC AGA GAG G -3′; NNMT (pBABE): forward, 5′-ctc tag gcg ccg gcc gga tcc ATG GAA TCA GGC TTC ACC TCC-3′, and reverse, 5′- ggt ctt ctc gtc cat gaa ttc CAG GGG TCT GCT CAG CTT CC -3′.

**Western Blot analysis**. Cultured cells or cells from human ccRCC and matched normal tissues were lysed with 0.5% NP-40 buffer containing 50 mM Tris-HCl (pH 7.5), 150 mM NaCl, 0.5% Nonidet P-40, and a mixture of protease inhibitors (Sigma-Aldrich). After centrifugation at $13,800 \times g$ and 4 °C for 15 min, supernatants were collected for western blotting according to standard procedures. Antibodies against DNA-PKcs (#38168, 1: 1000), phospho-Ser2056 DNA-PKcs (#68716, 1: 1000), $H_2AX$ (#7631, 1: 1000), γ-$H_2AX$ (#9718, 1: 1000), p53 (#9282, 1: 1000), phospho-Ser15 p53 (#9284, 1: 1000), KU70 (#4103, 1: 1000), and KU80 (#2753, 1: 1000) were purchased from Cell Signaling Technology. Antibody against NNMT was purchased from Abcam (#ab58743, 1: 1000). Antibody against Actin was purchased from Genscript (#A00702, 1: 800). Anti-K-Hcy antibody (1: 1000) was generated as described previously[56]. Chemiluminescence was measured on a Typhoon FLA 9500 instrument (GE Healthcare).

**IHC**. We collected 12 samples (beyond 232 paired samples) for NNMT IHC validation additionally. Sections of ccRCC and adjacent tissues were obtained from formalin-fixed, paraffin-embedded tissue blocks (not enrolled in the

proteogenomic cohort). Immunostaining was carried out as reported previously[97,98]. Sections were stained using relevant antibodies and the Envision detection kit (Dako). Immunostaining was quantified based on the number of immunoreactive cells (quantity score) and the staining intensity (intensity score), as reported[97,98].

**Metabolite quantification**. Human tissues were homogenized in ice-cold phosphate-buffered saline (PBS) and centrifuged, and supernatants were collected for Hcy quantification. Hcy concentrations were determined using an Axis Homocysteine Enzyme Immunoassay Kit (Axis-Shield). To assay metabolite levels, cells were harvested by PBS washing and denatured in pre-chilled 60% methanol (in ddH$_2$O, pre-cooled at $-80\,°C$ for 1–2 h). Cell lysates were centrifuged ($10,000 \times g$) at 4 °C for 5 min. Supernatants were vacuum-dried, re-dissolved in ddH$_2$O, and subjected to ultrafiltration on a polyvinylidene fluoride low protein binding membrane (Millex-GV4 and Millex-HV4, Millipore). Metabolites were extracted and Hcy was analyzed using LC-MS. SAM and SAH levels were detected using a SAM & SAH ELISA Combo Kit (Cell Biolabs). 1-Methylnicotinamide was measured using a UHPLC-QTOF-MS System (Agilent Technologies, 1290 LC, 6550 MS) as described previously[99]. Each assay was repeated in triplicate, and means were used for analysis.

**Lysine-homocysteinylation site identification in ccRCC tissues**. To identify lysine-homocysteinylation sites in tissue samples, ccRCC tumor and non-tumor tissues were ground in 0.5% NP-40 buffer, and supernatants were immunoprecipitated with anti-DNA-PKcs antibody and digested with trypsin. LC-MS/MS experiments were conducted on an EASY-nLC100 chromatograph coupled with an Orbitrap Elite (both from Thermo Fischer Scientific) equipped with an online nano-electrospray ion source. Peptides were desalted and suspended in 10 μL solvent A (solvent A: water with 0.1% formic acid; solvent B: acetonitrile with 0.1% formic acid). Each sample was loaded onto a self-packed C18 column (100 μm × 2 cm, 5 μm particle size), with a flow rate of 5 μL/min for 5 min and subsequently separated on the analytical column (C18, 75 μm × 20 cm) with a linear gradient from 5% solvent B to 90% over 120 min. The column was re-equilibrated at initial conditions for 15 min. The column flow rate was maintained at 200 nL/min. The mass spectrometer was set as follows: ion-transfer capillary, 275 °C; spray voltage, 2 kV; and full MS range, 400–2,000 m/z. Full mass spectra were acquired at 60,000 resolution with a target ion setting of $10^6$. One full MS scan was followed by 15 MS/MS scans, and multistage activation was enabled. The dynamic exclusion function was set as follows: repeat count, 2; repeat duration, 30 s; and exclusion duration, 60 s.

**DNA-PKcs in vitro kinase assay**. In vitro DNA-PKcs kinase assays were conducted as described previously[58]. In brief, 200 ng DNA-PKcs and 3 μg p53 were incubated in a buffer containing 50 mM HEPES (pH 7.4), 100 mM KCl, 10 mM MgCl$_2$, 2 mM EGTA, 0.1 mM EDTA, and 1 mM ATP at 30 °C for 30 min. Y-shape DNA and KU70/KU80 were added as indicated. Reactions were terminated by addition of sodium dodecyl sulfate (SDS) sample loading buffer and boiling for 5 min. Samples were subjected to SDS-polyacrylamide gel electrophoresis and immunoblotting using site-specific antibody against p53.

**DNA-PK kinase assay**. DNA-PKcs activity was measured using the ADP-Glo$^{TM}$ + DNA-PK kinase system (Promega, Cat#4107). Briefly, we isolated DNA-PKcs protein from cells subjected to various treatments. To measure DNA-PKcs activity, 1 μL 5% DMSO, 2 μL of enzyme, and 2 μL of substrate/ATP mix were added to the wells of a 384-well plate. The plate was incubated at room temperature for 60 min. Then, 5 μL of ADP-Glo$^{TM}$ reagent was added and the plate was incubated at room temperature for 40 min. Consequently, 10 μL of kinase detection reagent was added and the plate was incubated at room temperature for 30 min. Luminescence was recorded with an integration time of 0.5–1 s.

**Cell proliferation assay**. Cell proliferation was assessed using the Cell Counting Kit-8 (Dojindo Laboratories). In brief, cells were seeded in a 96-well plate at $4 \times 10^3$ cells/well and allowed to adhere. Cell Counting Kit-8 solution (10 μL) was added to each well, and the cells were incubated in 5% CO$_2$ at 37 °C for 2 h. Cell proliferation was determined by measuring the absorbance at 450 nm.

**Comet assay**. A Comet Assay Kit (Trevigen) was used to detect single- and double-stranded DNA breaks in cultured cells and tissues. Slides were examined under a Leica DMI 4000B epifluorescence microscope (425–500-nm excitation). Comet slides were used for each condition. In normal cells, fluorescence is mostly confined to the nucleus because intact DNA cannot migrate. In DNA-damaged cells, DNA is denatured with an alkaline or neutral solution to detect single- or double-stranded breaks, respectively; negatively charged DNA fragments are released from the nucleus and migrate toward the anode.

**In Vivo Xenograft studies**. Four-to-six-week-old Balb/C nude male mice were obtained from Shanghai SLAC Laboratory Animal Co., Ltd. All mice were housed on a 12 h light/dark cycle at 25 °C. Control and NNMT-overexpressing ACHN and

786-O cell lines were subcutaneously transplanted into the left and right flanks of each mouse. For the IR group, irradiated control and NNMT-overexpressing cells were transplanted into the left and right flanks of each mouse. For the IR + NAC group, irradiated control and NNMT-overexpressing cells were transplanted into the left and right flanks of each mouse, and the mice were intraperitoneally injected with NAC (500 mg/kg) every other day. This study is under the guidelines of the Institutional Animal Care and Use Committee (IACUC), Fudan University. The maximal permitted tumor size is 20 mm in an average diameter for mice, in accordance with guidelines of IACUC. At the end of the experiment, following euthanasia, tumors were excised, weighed, and imaged. All procedures were approved by IACUC, Fudan University. Ethical review approval number 201802143 S was obtained from the Department of experimental animal science, Fudan University.

**Quantification and statistical analysis**. Quantification methods and statistical analysis methods for proteomic and integrated analyses were mainly described and referenced in the respective Method Details subsections.

Additionally, standard statistical tests were used to analyze the clinical data, including but not limited to Student's t test, Fisher's exact test, Kruskal–Wallis test, log-rank test. Statistical significance was considered when p value < 0.05. To account for multiple-testing, the p values were adjusted using the Benjamini–Hochberg FDR correction. Kaplan–Meier plots (two-sided log-rank test) were used to describe OS and PFS. Variables associated with overall survival were identified using univariate Cox proportional hazards regression models. Significant factors in univariate analysis were further subjected to a multivariate Cox regression analysis. All the analyses of proteogenomic data were performed in R and GraphPad Prism. For functional experiments, each was repeated at least three times independently.

**Reporting summary**. Further information on research design is available in the Nature Research Reporting Summary linked to this article.

## Data availability

Proteome raw data have been deposited to the ProteomeXchange Consortium (dataset identifier: PXD030344) via the iProX partner repository (https://www.iprox.cn/)[100] under Project ID: IPX0001962000. WES data files were deposited to NODE (https://www.biosino.org/node) under Project ID: OEP000796 and the European Genome-phenome Archive (EGA) under project ID EGAD00001008556. Data is available upon request through EGA without any restrictions, and will be available permanently. EGA access can be gained by contacting Jinwen Feng (jinwenf@fudan.edu.cn). Proteome quantification matrix and H&E-stained section images of this study were deposited in Figshare (https://doi.org/10.6084/m9.figshare.17206589). TCGA ccRCC data were downloaded from Xena (https://xenabrowser.net/). CPTAC ccRCC could be accessed at https://cptac-data-portal.georgetown.edu/study-summary/S044 and http://ccrcc.cptac-data-view.org/. GDSC and CCLE data could be accessed DepMap data portal (https://depmap.org/portal/). Source data are provided with this paper.

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

## Acknowledgements

This work is supported by National Program on Key Basic Research Project (2019YFA0801900 [C.D.], 2019YFC1316000 [C.D.]); National Key R&D Program of China (2017YFA0505102 [C.D.], 2016YFA0502500 [C.D.], 2018YFE0201603 [C.D.], 2017YFA0505101 [C.D.], 2020YFE0201600 [C.D.], 2019YFC1316005 [H.L.Z.]); National Natural Science Foundation of China (31770886 [C.D.], 31972933 [C.D.], 81802525 [Y.Y.Q], 82172817 [Y.Y.Q], 81872099 [D.W.Y.], 82172741[D.W.Y.]); Natural Science Foundation of Shanghai (20ZR1413100 [H.L.Z.]); Shanghai Municipal Science and Technology Major Project (2017SHZDZX01 [C.D.]); Major Project of Special Development Funds of Zhangjiang National Independent innovation Demonstration Zone (ZJ2019-ZD-004 [C.D.]); China Postdoctoral Science Foundation (2020T130114 [J.W.F.]); Fudan original research personalized support project (C.D.), and Shanghai "Rising Stars of Medical Talent" Youth Medical Talents–Specialist Program (Y.Y.Q). We thank the Lifehealthcare Clinical Laboratories (Hangzhou, China) for whole exome sequencing service and related technical assistance.

## Author contributions

Y.Y.Q., H.L.Z., J.Y.Z., D.W.Y., and C.D. conceived and planned the project. Y.Y.Q., G.H.S., H.K.W., D.L.C. H.L.G., and M.H.S. were responsible for sample and clinical information collection. L.L.Z., X.P.C., G.J.Y., J.C.L., H.X., L.L.L., and S.B.T. contributed to sample preparation. Y.Y.Q., W.H.X., F.J.X., and X.Z. planned and carried out the validation experiments. Y.Y.Q., J.W.F., X.H.W., L.B., Y.L., G.J.Y., J.C.L., and J.G.Q. analyzed the data and contributed to the interpretation of the results. C.D. took the lead in writing the manuscript. All authors provided critical feedback and helped shape the research, analysis, and manuscript.

## Competing interests

The authors declare no competing interests.
