## [Peer Review File · Nature Communications]

A proteogenomic analysis of clear cell renal cell carcinoma in a Chinese populationEditorial Note: This manuscript has been previously reviewed at another journal that is not operating a transparent peer review scheme. This document only contains reviewer comments and rebuttal letters for versions considered at *Nature Communications*. Mentions of prior referee reports have been redacted.

REVIEWERS' COMMENTS

Reviewer #1 (Remarks to the Author):

The manuscript by Qu and colleagues titled “A Proteogenomic Atlas of Clear Cell Renal Cell Carcinoma in a Chinese Population” is an interesting proteogenomics (mainly proteomics) investigation of a relatively large cohort of 232 renal cell carcinoma samples, with matching adjacent normal. This is a resubmission of a manuscript previously under review at [REDACTED] and the authors have carefully replied to previous comments. The authors have also modified several of their previous analysis. While the paper is still dense, which is the case for most proteogenomics papers, this manuscript is without a doubt in-line with several recently published proteogenomics manuscripts in the literature. What is unique about this manuscript is the relatively large cohort of samples (larger than all other proteogenomics papers that I am aware of) and the use of a unique population cohort. This is not simply a redo of a CPTAC paper, the current study provides >200 patients of Chinese ancestry, which should provide interesting comparisons to the mainly Caucasian (western) cohorts analyzed by other consortia. The proteomics data quality is rigorous and several of the higher-level analysis provide interesting insights for follow-up in the future (again like all other proteogenomics papers in the literature). I only have several minor comments the authors should consider:

- 1) Page 9; line 169: the eight orders of magnitude is most likely an artifact of the quantitative method used here. I suggest toning this statement down a bit.
- 2) Page 14; lines 291-292: How were these 20 proteins mentioned assigned as “plasma proteins”. While some of them may have been detected in plasma, they don’t strike me a classic plasma proteins?
- 3) In response to one of my previous comments the authors now cite all CPTAC papers. This is not what I asked, and I think it’s misleading and biased. There are several other studies published in some of the top journals utilizing proteogenomics technologies. Readers of this paper should not get the impression the CPTAC is an acronym for proteogenomics. The authors should do some literature searches and reference the literature more unbiased.

Reviewer #2 (Remarks to the Author):

satisfied with the authors response to the reviewers

Reviewer #3 (Remarks to the Author):

The authors have carefully addressed all the reviewers comments, and have improved the clarity of the manuscript. Even though some weaknesses remain, most notably that RNA-seq data is missing for most tumors, the data set will be a very valuable resource for the community, and therefore, I recommend publication of the manuscript.

Reviewer #1 (Remarks to the Author):

The manuscript by Qu and colleagues titled “A Proteogenomic Atlas of Clear Cell Renal Cell Carcinoma in a Chinese Population” is an interesting proteogenomics (mainly proteomics) investigation of a relatively large cohort of 232 renal cell carcinoma samples, with matching adjacent normal. This is a resubmission of a manuscript previously under review at [REDACTED] and the authors have carefully replied to previous comments. The authors have also modified several of their previous analysis. While the paper is still dense, which is the case for most proteogenomics papers, this manuscript is without a doubt in-line with several recently published proteogenomics manuscripts in the literature. What is unique about this manuscript is the relatively large cohort of samples (larger than all other proteogenomics papers that I am aware of) and the use of a unique population cohort. This is not simply a redo of a CPTAC paper, the current study provides >200 patients of Chinese ancestry, which should provide interesting comparisons to the mainly Caucasian (western) cohorts analyzed by other consortia. The proteomics data quality is rigorous and several of the higher-level analysis provide interesting insights for follow-up in the future (again like all other proteogenomics papers in the literature). I only have several minor comments the authors should consider:

1) Page 9; line 169: the eight orders of magnitude is most likely an artifact of the quantitative method used here. I suggest toning this statement down a bit.

Response:

Thanks for reviewer’s comment. The reviewer is correct that the eight orders of magnitude is based on the quantification methods. Following reviewer’s comments, we tone down the statement by deleting the sentence “The dynamic range of proteins detected spanned eight orders of magnitude.” in the revised manuscript.

2) Page 14; lines 291-292: How were these 20 proteins mentioned assigned as “plasma proteins”. While some of them may have been detected in plasma, they don’t strike me a classic plasma proteins?

Response:

We thank the reviewer for the comments. The reviewer is correct that these 20 proteins can be

detected in plasma but are not classic plasma proteins. The annotations about these 20 proteins were derived from the human protein atlas (HPA) database (<https://www.proteinatlas.org/>), and were further confirmed in the Plasma Proteome Database (<http://plasmaproteomedatabase.org/>). In our manuscript, these “plasma proteins” were used to predict patients with short progression free survival (PFS) among early stage ccRCC patients (**Figure RL 1**). As these proteins could be detected in plasma, it has potential in clinical applications. We apologized for the unclear annotation about these 20 proteins and revised it to a more appropriate description “proteins that could be detected in plasma”. Specifically, we revised the sentence as “In addition, among proteins significantly upregulated in short PFS group ($FC > 2$, $p < 0.05$), 20 proteins which could be detected in plasma (annotated by HPA database; Uhlén M, et al. Science. 2015. PMID: 25613900) showed significantly correlation with clinical outcomes. We used these 20 proteins to distinguish between short and long PFS patients which achieved a high accuracy, with the area under the receiver operating characteristic (AUROC) of 0.87.” in the revision.

Figure RL 1

A, Among proteins significantly upregulated in short PFS group ($FC > 2$, two-sided t test, $p < 0.05$), 20 proteins could be detected in plasma. Higher expressions of these proteins were associated with shorter PFS.

B, The area under the receiver operating characteristic (AUROC) of the 20 proteins predictor.

3) In response to one of my previous comments the authors now cite all CPTAC papers. This is not what I asked, and I think it's misleading and biased. There are several other studies published in some of the top journals utilizing proteogenomics technologies. Readers of this paper should

not get the impression the CPTAC is an acronym for proteogenomics. The authors should do some literature searches and reference the literature more unbiased.

Response: Thanks again for the comments. We apologized for misunderstanding of previous comments from the reviewer. The reviewer is true that besides CPTAC studies, there are other excellent tumor proteogenomics studies (Ge S, et al. Nat Commun. 2018. PMID: 29520031; Jiang Y, et al. Nature. 2019. PMID: 30814741; Johansson HJ, et al. Nat Commun. 2019. PMID: 30962452; Li C, et al. Cancer Cell. 2020. PMID: 32888432; Xu JY, et al. Cell. 2020. PMID: 32649877; Nassiri F, et al. Nature. 2021. PMID: 34433969). We cited these studies in the revised manuscript. We also adjusted the references to CPTAC to make the citation more accurate.

Reviewer #2 (Remarks to the Author):

satisfied with the authors response to the reviewers

Response:

Thank you for your positive comments and suggestions!

Reviewer #3 (Remarks to the Author):

The authors have carefully addressed all the reviewers comments, and have improved the clarity of the manuscript. Even though some weaknesses remain, most notably that RNA-seq data is missing for most tumors, the data set will be a very valuable resource for the community, and therefore, I recommend publication of the manuscript.

Response:

Thank you for your positive comments and suggestions!